# Dissociable neural substrates of integration and segregation in exogenous attention

Yujie Chen[1†], Ai-Su Li[1†], Yang Yu[2], Su Hu[2], Xun He[3*], Yang Zhang[1*]

[1]Department of Psychology, Soochow University, Suzhou, China; [2]Department of Radiology, The First Affiliated Hospital of Soochow University, Suzhou, China; [3]School of Psychology, Bournemouth University, Poole, United Kingdom

*For correspondence:
xhe@bournemouth.ac.uk (XH);
yzhangpsy@suda.edu.cn (YZ)

[†]These authors contributed equally to this work

Competing interest: The authors declare that no competing interests exist.

## eLife Assessment

This **important** study uses an optimized IOR-Stroop fMRI paradigm to dissociate integration and segregation processes and to show that attentional orienting modulates conflict processing at both the semantic and response levels. The evidence is **compelling**, supporting the integration-segregation theory of exogenous attention in inhibition of return while also deepening our understanding of how attentional orienting shapes downstream cognitive processing. The work will therefore be of broad interest to researchers in attention and cognitive control.

**Abstract** The integration-segregation theory proposes that early facilitation and later inhibition (i.e. inhibition of return [IOR]) in exogenous attention arises from the competition between cue-target event integration and segregation. Although widely supported behaviorally, the theory lacked direct neural evidence. Here, we used event-related functional magnetic resonance imaging (fMRI) in human participants with an optimized cue-target paradigm to test this account. Cued targets elicited stronger activation in the frontoparietal attention networks, including the bilateral frontal eye field (FEF), intraparietal sulcus (IPS), right temporoparietal junction (TPJ), and left dorsal anterior cingulate cortex (dACC), consistent with the notion of attentional demand of reactivating the cue-initiated representations for integration. In contrast, uncued targets engaged the medial temporal cortex, particularly the bilateral parahippocampal gyrus (PHG) and superior temporal gyrus (STG), reflecting the segregation processes associated with new object-file creation and novelty encoding. These dissociable activations provide the first direct neuroimaging evidence for the integration-segregation theory. Moreover, we observed neural interactions between IOR and cognitive conflict, suggesting a potential modulation of conflict processing by attentional orienting. Taken together, these findings provide new insights into exogenous attention by clarifying the neural underpinnings of integration and segregation and uncovering the interaction between spatial orienting and conflict processing.

## Introduction

Salient visual stimuli, such as abrupt onsets, involuntarily capture attention. This process, known as exogenous or reflexive attentional orienting, is crucial for efficient visual search in cluttered scenes (*Klein et al., 2023*; *Li et al., 2023*; *Ma et al., 2011*; *Wang and Klein, 2010*; *Wolfe and Horowitz, 2017*). Studies employing the cue-target paradigm (*Posner and Cohen, 1984*) have shown that the exogenous attentional orienting has a biphasic temporal pattern. An uninformative peripheral cue initially facilitates subsequent target processing at the cued location at short stimulus-onset

asynchronies (SOAs), and later turns into inhibiting responses at the cued location at long SOAs (typically over 200 ms), a phenomenon known as inhibition of return (IOR) (*Klein, 2000*; *Lupianez et al., 2006*; *Posner et al., 1985*; *Seidel Malkinson et al., 2024*). This characteristic shift from facilitation to inhibition has fueled decades of theoretical debate, giving rise to multiple competing accounts of its underlying mechanisms (*Funes et al., 2008*; *Klein and Dick, 2002*; *Lupiáñez, 2010*; *Lupiáñez et al., 2001*; *Milliken et al., 2000*; *Prime and Jolicoeur, 2009*; *Taylor and Klein, 1998*; *Vivas et al., 2007*).

## The integration-segregation theory

One of the most influential and extensively developed theories explaining the biphasic effect is the integration-segregation theory proposed by *Funes et al., 2008*; *Lupiáñez et al., 2001*; *Milliken et al., 2000*. Rooted in the object file framework (*Kahneman et al., 1992*), this theory attributes the biphasic pattern to the dynamic competition between cue-target integration and segregation. The integration process favors integrating the targets at the cued locations into an existing episodic representation (an object file) that has been activated by the preceding peripheral cue (*Kahneman et al., 1992*), whereas the segregation process tends to create a new episodic representation for targets at the uncued locations (*Funes et al., 2008*; *Lupiáñez and Jesús Funes, 2005*; *Lupiáñez et al., 2001*; *Milliken et al., 2000*). At short SOAs (e.g. less than 200 ms), integration on the cued trials is more efficient than segregation on the uncued trials, resulting in faster responses for the cued than the uncued trials (i.e. the facilitation effect). At longer SOAs, however, the original cue-activated object file gradually closed, making it less efficient to integrate new stimuli. Consequently, constructing a new object file at the uncued location gradually becomes easier than updating the closing one, resulting in the IOR effect.

Over the past two decades, the integration-segregation theory has been widely accepted as a flexible and extensible framework for explaining the accumulating IOR research findings (*Chen et al., 2007*; *Funes et al., 2008*; *Hu et al., 2011*; *Li et al., 2018*; *Luo et al., 2010*; *Lupiáñez et al., 2001*; *Lupiáñez et al., 2007*; *Zu et al., 2023*). Originally developed to explain spatial IOR (*Funes et al., 2008*; *Lupiáñez and Jesús Funes, 2005*; *Lupiáñez et al., 2001*; *Milliken et al., 2000*), it has subsequently been extended to non-spatial forms of IOR (e.g. color-, shape-, or frequency-based IOR) in both visual and auditory modalities (*Chen et al., 2007*; *Hu et al., 2011*). When targets and cues share both non-spatial and spatial features, these consecutive stimuli are integrated into a single event representation, hindering the detection of the target (*Chen et al., 2007*; *Hu et al., 2011*). More recently, such integrative interference has also been observed in cross-modal IOR, revealing supramodal mechanisms involving abstract semantic features (*Zu et al., 2023*). Researchers have further extended the explanatory scope of the integration-segregation framework to electrophysiological data (*Li et al., 2018*; *Martín-Arévalo et al., 2014*; *Martín-Arévalo et al., 2016*). For example, enhanced P3 amplitudes at the uncued locations have been interpreted as reflecting greater cognitive demands associated with new object-file creation underlying IOR (*Li et al., 2018*), whereas reduced P1 amplitudes at the cued locations have been taken to index a perceptual detection cost arising from disrupted cue-target integration (*Martín-Arévalo et al., 2014*; *Martín-Arévalo et al., 2016*).

Furthermore, the theory also succeeds in accounting for task-dependent variations in IOR that are difficult to explain by other attentional theories (*Chen et al., 2007*; *Lupiáñez and Milliken, 1999*; *Lupiáñez et al., 2001*; *Lupiáñez et al., 2007*). By positing that task demands modulate the timing of object-file closure, the framework predicts an earlier IOR onset in detection tasks (favoring an early file closure for new event encoding) than that in discrimination tasks (i.e. late closure due to information accumulation) (*Lupiáñez and Milliken, 1999*; *Lupiáñez et al., 2001*). Crucially, *Lupiáñez et al., 2007*, found that, at long SOAs, frequent targets elicited the expected IOR, whereas infrequent targets instead received facilitation. This dissociation cannot be explained by cue-driven attentional capture and disengagement accounts alone, and instead suggests a task-dependent cue-target integration (*Lupiáñez, 2010*). Similarly, in auditory attention, *Chen et al., 2007*, found that task-irrelevant features can either enhance or eliminate the IOR effect depending on whether the cue and target share the same task-relevant dimension, a pattern considered to be better explained by the integration-segregation theory than by the traditional accounts. Collectively, by emphasizing the cue-target interplay, the integration-segregation account provides a unified theoretical framework of exogenous attention accommodating diverse stimulus features, modalities, and task demands.

## The challenge of neural verification

Despite the strength, current support for the integration-segregation theory remains largely inferential, due to the fact that the recently hypothesized dual processes of integration and segregation had not been directly evidenced in brain activities. Operating under the assumption that IOR reflects a time-dependent inhibitory state that builds up with the increase of SOA (*Lepsien and Pollmann, 2002*; *Mayer et al., 2004b*), the past studies typically contrasted long vs. short SOAs to capture the neural dynamics underlying the inhibitory phase of visual attentional orienting. This contrast was typically examined either by collapsing across cued and uncued trials (*Lepsien and Pollmann, 2002*; *Mayer et al., 2004a*; *Müller and Kleinschmidt, 2007*; *Zhou and Chen, 2008*) or by analyzing the cued and uncued conditions separately without directly comparing them (*Mayer et al., 2004b*). These studies observed the involvement of the frontoparietal attention network, particularly the frontal eye fields (FEF), anterior cingulate cortex (ACC), and inferior parietal lobule (IPL). However, these SOA-based contrasts were insufficient for testing the integration-segregation framework, as they only captured the temporal dynamics in attentional orienting and in the process missed the functional distinction between integration and segregation that characterizes the theory.

To directly test the functional distinction of event integration (for the cued targets) vs. segregation (for the uncued targets), it is necessary to compare the cued and uncued conditions. Studies that attempted this direct comparison have yielded mixed findings. While *Chen et al., 2006*, identified a cue-validity effect during the inhibitory period, this effect was confined to the left FEF, and other studies did not observe significant neural differences between the cued and uncued trials during the inhibitory period (*Lepsien and Pollmann, 2002*; *Mayer et al., 2004b*). Given that the integration-segregation account predicts distinct neural processing for cued and uncued targets, clear evidence for such a dissociation during inhibition remains limited. This limitation may reflect statistical power constraints inherent in event-related functional magnetic resonance imaging (ER-fMRI) experiments (despite their high psychological validity, i.e. estimation efficiency), further aggravated by the suboptimal temporal structure of stimulus sequences, and the limited sample sizes and trial numbers (*Buracas and Boynton, 2002*; *Liu, 2004*; *Liu and Frank, 2004*; *Liu et al., 2001*; *Wager and Nichols, 2003*).

## Overview of the present study

To obtain direct neuroimaging evidence for the integration-segregation theory, the present study employed ER-fMRI with a stimulus sequence optimized with a genetic algorithm (GA) (*Wager and Nichols, 2003*). This flexible optimization approach was adopted to maximize the statistical power of contrast detection while maintaining a high estimation efficiency of the hemodynamic response function (HRF), thereby addressing the power limitations that had negatively impacted previous neuroimaging IOR studies. Guided by the integration-segregation framework, we predicted dissociable neural signatures for the cued vs. uncued targets corresponding to their divergent processing requirements. Specifically, targets at the cued locations would engage an integration process to update the existing file, recruiting regions associated with information integration and attentional reorienting such as the FEF (*Astafiev et al., 2003*; *Corbetta and Shulman, 2002*; *Liu et al., 2023*). In contrast, targets appearing at the uncued locations would engage a segregation process to establish a new object file, presumably recruiting regions involved in new episodic encoding, such as the parahippocampal gyrus (PHG) (*Burgess et al., 2002*; *Danieli et al., 2023*; *Hayes et al., 2007*; *Li et al., 2016*; *Menon et al., 2000*; *Torres-Morales and Cansino, 2024*).

Beyond the primary focus on the neural mechanisms of the integration-segregation framework (i.e. the IOR generation mechanisms), the current study also employed the experimental design to examine how IOR modulates ongoing cognitive processing (i.e. the IOR expression mechanisms). Specifically, we embedded a modified Stroop task within the cue-target paradigm to systematically manipulate cognitive conflict at the target-processing stage (*De Houwer, 2003*; *van Veen and Carter, 2005*; *van Veen et al., 2001*). These conflict types were operationalized through three distinct stimulus types: a neutral condition (non-color words shown in color, producing no conflict), a semantic conflict condition (word meaning and ink color were incongruent, but mapped to the same response), and a combined semantic-response conflict condition (word meaning and ink color were mismatched and mapped to different responses). This manipulation allowed us to examine how spatial attention interacts with distinct levels of cognitive conflict. A previous fMRI study by *Chen et al., 2006*, reported dissociable neural signatures for the semantic and response conflicts when spatial attention

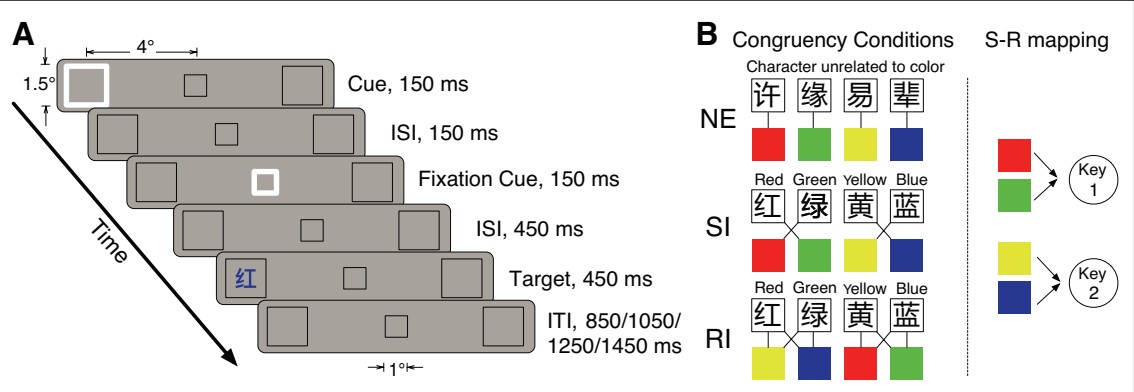

**Figure 1.** Experimental materials. (**A**) Trial sequence and display sizes. Each trial started with a 150 ms non-informative cue presented at one of the two peripheral boxes. After a 150 ms interstimulus interval (ISI), a 150 ms fixation cue was presented at the central fixation box. Following a further 450 ms ISI, the target, a colored Chinese character, appeared at one of the two target locations with equal probabilities and remained on the screen for 450 ms. The trial ended with a variable intertrial interval (ITI) of 850, 1050, 1250, or 1450 ms (with equal probabilities). (**B**) The character-color combinations in the three congruency conditions. In the neutral condition (first row), the characters were not color-related. In the other conditions, the characters were color names (translation added for illustration purposes). S-R mapping = stimulus-response mapping; NE = neutral; SI = semantically incongruent; RI = response-incongruent.

was engaged. However, that study manipulated response eligibility by excluding certain incongruent color words from the response set (*Milham et al., 2001*). This design choice has been criticized for conflating the semantic and response conflicts, as ineligible distractors may not be processed in the same way as response-relevant words (*van Veen and Carter, 2005*). To address this limitation, the present study adopted a refined Stroop design that clearly separates the semantic and response conflicts while keeping all stimuli response-relevant (*De Houwer, 2003*; *van Veen and Carter, 2005*; *van Veen et al., 2001*), providing a more precise test of how spatial attention modulates these two distinct conflict types in brain activities.

## Results

Participants performed a spatial cueing task (long SOA to elicit IOR) combined with a Stroop paradigm adapted for colored Chinese characters (*Chen et al., 2006*; *Figure 1A*), with the characters appearing at either the cued or the uncued location. The experimental manipulation dissociated the semantic and response conflicts, following a well-established three-condition design (*Figure 1B*). These conditions were neutral (NE; non-color characters), semantically incongruent (SI; the character and the color are incongruent but mapped to the same response, causing only the semantic conflict), and response-incongruent (RI; the character and the color are incongruent and mapped to opposite responses, causing both the semantic and response conflicts) (*De Houwer, 2003*; *van Veen and Carter, 2005*). Participants responded using two keys, each assigned to two colors. To ensure sufficient statistical power for detecting condition-specific neural differences, the ER-fMRI design was optimized using the GA (*Wager and Nichols, 2003*), with the goal of maximizing experimental efficiency for three contrasts of interest. To directly evaluate the prediction of the integration-segregation theory, we first examined the brain activity differences between the conditions of cued-NE (targets at the cued location in the neutral condition) and uncued-NE (targets at the uncued location in the neutral condition). Subsequent analyses examined how IOR modulated the conflict-related neural activities. The contrast of cued-SI minus cued-NE vs. uncued-SI minus uncued-NE was used to assess the effect of IOR on semantic conflict processing, whereas the contrast of cued-RI minus cued-SI vs. uncued-RI minus uncued-SI was employed to capture the modulation of response conflict by IOR.

### Behavioral results

Mean reaction times (RTs) and accuracies are shown in *Figure 2*. A two-way (cue validity × congruency) repeated-measures analysis of variance (rm-ANOVA) for the RTs revealed a significant IOR effect (main effect of cue validity), $F(1, 28) = 12.057$, p = 0.002, $\eta_p^2 = 0.301$, showing slower responses to targets

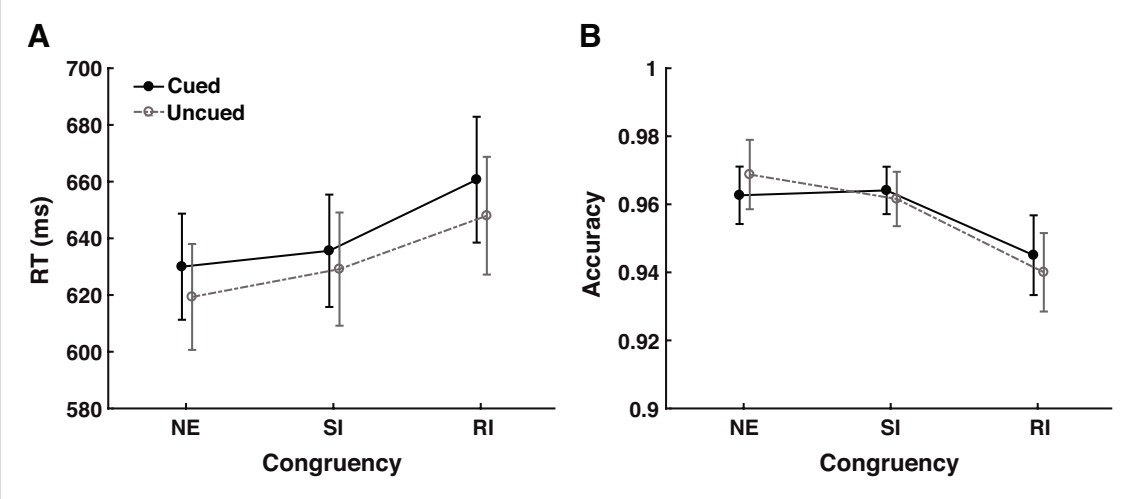

**Figure 2.** Behavioral results. Mean reaction times (**A**) and accuracies (**B**) as a function of cue validity and congruency. NE = neutral; SI = semantically incongruent; RI = response-incongruent. Error bars extend to one standard error of the mean (SEM), N = 29.

at the cued location (M = 642 ms, SE = 20 ms) than at the uncued location (M = 632 ms, SE = 19 ms). The main effect of congruency was also significant, $F(1.53, 42.88) = 29.602$, p < 0.001, $\eta_p^2 = 0.514$ (the Greenhouse-Geisser correction was applied due to the violation of the sphericity assumption). Post hoc comparisons with the Holm-Bonferroni correction (*Holm, 1979*) revealed significant differences among all conditions (NE vs. RI: $t(28) = -2.179$, p = 0.038, Cohen's $d = 0.071$; NE vs. SI: $t(28) = -5.957$, p < 0.001, Cohen's $d = 0.275$; SI vs. RI: $t(28) = -6.715$, p < 0.001, Cohen's $d = 0.203$), with NE showing the shortest RT (M = 624 ms, SE = 19 ms), followed by SI (M = 632 ms, SE = 20 ms), then RI (M = 654 ms, SE = 21 ms). These data demonstrated typical Stroop interference effects (*van Veen and Carter, 2005*) in both the semantic (SI-NE difference) and response conflicts (RI-SI difference). The interaction between cue validity and congruency did not approach significance, $F(2, 56) = 0.930$, p = 0.401, $\eta_p^2 = 0.032$. To further investigate whether cue validity modulated the two conflict components, we conducted planned analyses examining its interactions with semantic conflict (SI vs. NE) and response conflict (RI vs. SI) (*Chen et al., 2006*). The results showed that cue validity did not significantly interact with either semantic conflict ($F(1, 28) = 0.968$, p = 0.334, $\eta_p^2 = 0.033$) or response conflict ($F(1, 28) = 1.502$, p = 0.231, $\eta_p^2 = 0.051$).

The rm-ANOVA for the accuracy data (*Figure 2B*) only showed a significant main effect of congruency, $F(2, 56) = 7.685$, p = 0.001, $\eta_p^2 = 0.215$. Post hoc comparisons confirmed that this came from a lower accuracy in the RI condition (M = 0.943, SE = 0.009) than in the NE (M = 0.966, SE = 0.009; $t(28) = -3.596$, p = 0.002, Cohen's $d = 0.446$) and SI (M = 0.963, SE = 0.009; $t(28) = -3.150$, p = 0.005, Cohen's $d = 0.391$) conditions. No significant difference was found between the NE and SI conditions ($t(28) = 0.446$, p = 0.657, Cohen's $d = 0.055$). The main effect of cue validity ($F(1, 28) = 0.021$, p = 0.887, $\eta_p^2 < 0.001$) and the interaction ($F(2, 56) = 1.298$, p = 0.281, $\eta_p^2 = 0.044$) were not significant. Given that accuracy did not decrease with faster response times, no speed-accuracy trade-off was noticed in the current data.

## Neuroimaging results

### IOR effect in the neutral condition

The contrast between the cued-NE and uncued-NE conditions was examined to identify the underlying neural mechanisms of the IOR effect during the processing of neutral targets. Whole-brain fMRI findings revealed two distinct activation patterns in response to these conditions (*Figure 3A*). Relative to the uncued-NE condition, the cued-NE condition showed enhanced activations in the dorsal attention network (DAN), including the bilateral FEF and intraparietal sulcus (IPS), along with the right-lateralized temporoparietal junction (TPJ) from the ventral attention network (VAN), and the left dorsal anterior cingulate cortex (dACC). In contrast, the uncued-NE condition demonstrated stronger activations in the bilateral PHG and superior temporal gyrus (STG) than the cued-NE condition. Notably, in

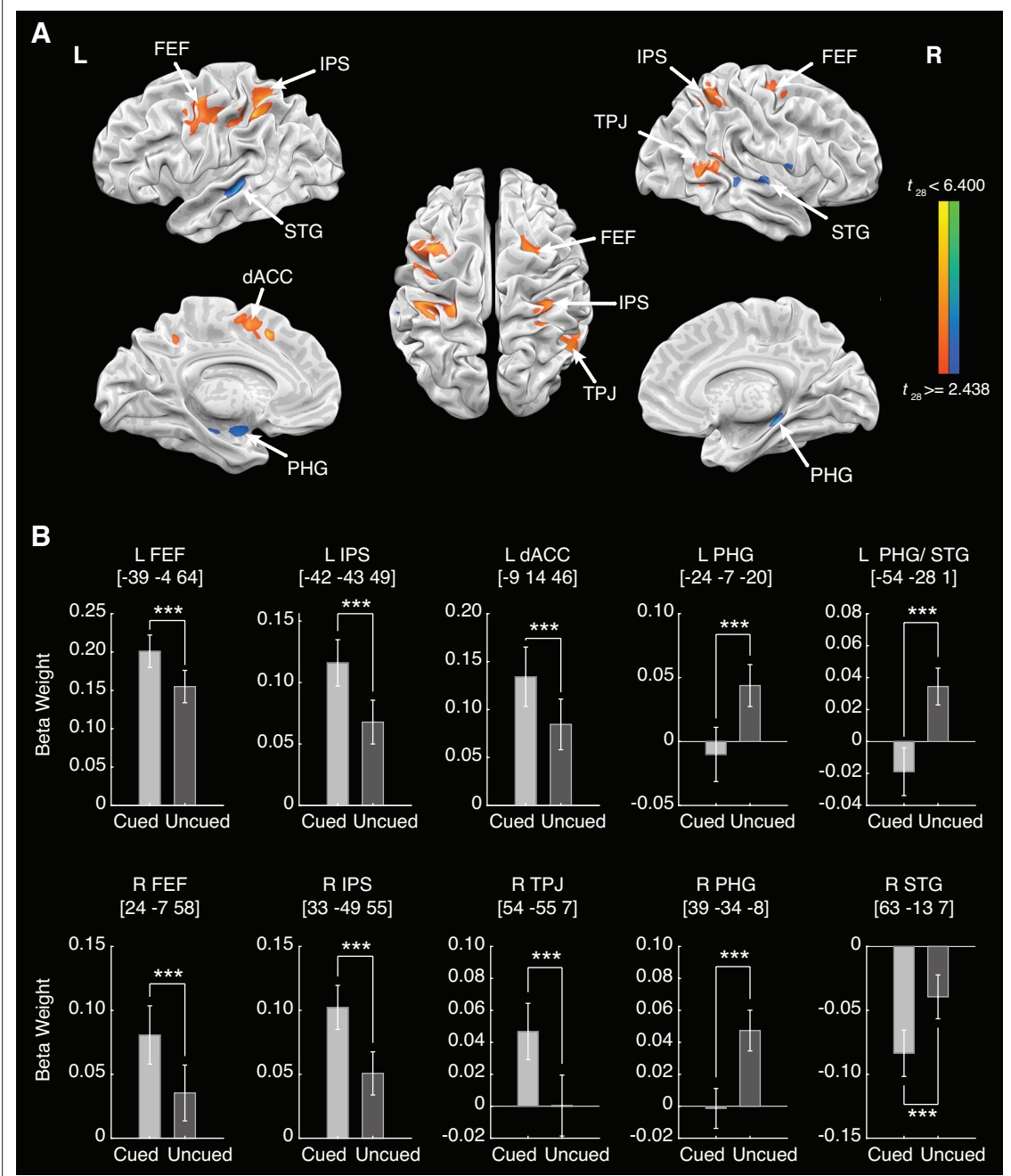

**Figure 3.** Inhibition of return (IOR) effect in the neutral condition and parameter estimation. (**A**) Brain regions showing significant activations in the contrast between the cued-neutral (NE) and uncued-NE conditions, with a threshold of p<0.005 (uncorrected) with a minimum cluster size of 540 mm³ (20 voxels), yielding a corrected p<0.05 based on 2500 Monte Carlo simulations in BrainVoyager. Warm colors represent stronger activations in the cued condition, and cold colors represent stronger activations in the uncued condition. (**B**) Parameter estimates for each activation region. Error bars extend to 1 SEM (N = 29). L = left; R = right. ***p<0.001.

the left hemisphere, these activations formed a continuous cluster spanning both regions (labeled as PHG/STG in *Table 1*). To further compare the activity levels in each brain region between the cued-NE and uncued-NE conditions, paired *t*-tests were conducted on the average parameter estimates (beta weights) in the left and right IPS, FEF, PHG, and STG, as well as the left dACC and right TPJ (all ps<0.001, see *Figure 3B*). Detailed information on the activated regions' coordinates, cluster sizes, and statistical significance is provided in *Table 1*.

**Table 1.** Brain regions showing significant activation differences between the cued-neutral (NE) and uncued-NE conditions.

| Regions | Laterality | Cluster (voxels) | MNI coordinates peak | | | $T_{max}$ | $T_{mean}$ | BA |
|---|---|---|---|---|---|---|---|---|
| | | | x | y | Z | | | |
| | L | 146 | −39 | −4 | 64 | 4.51 | 3.43 | 6 |
| Frontal eye field | R | 28 | 24 | −7 | 58 | 4.32 | 3.52 | 6 |
| | L | 120 | −42 | −43 | 49 | 4.73 | 3.57 | 40 |
| Intraparietal sulcus | R | 82 | 33 | −49 | 55 | 5.12 | 3.78 | 40 |
| Dorsal anterior cingulate cortex | L | 36 | −9 | 14 | 46 | 5.05 | 3.46 | 32 |
| Temporoparietal junction | R | 75 | 54 | −55 | 7 | 4.23 | 3.45 | 39 |
| | L | 22 | −24 | −7 | −20 | −4.11 | −3.46 | / |
| Parahippocampal gyrus | R | 44 | 39 | −34 | −8 | −5.44 | −3.83 | / |
| Parahippocampal gyrus/superior temporal gyrus | L | 84 | −54 | −28 | 1 | −4.42 | −3.49 | 21 |
| Superior temporal gyrus | R | 32 | 63 | −13 | 7 | −4.22 | −3.48 | 22 |

MNI = Montreal Neurological Institute; BA = Brodmann area; L = left; R = right. '/' indicates that no BA could be assigned.

## Effect of IOR on semantic conflict

Although the behavioral results did not reveal any significant modulation of IOR in the magnitude of either semantic conflict or response conflict, differential neural modulations were observed between these conditions (summarized in *Figure 4* and *Table 2*). The effect of IOR on the semantic conflict was examined as the contrast between the SI-NE differences (SI minus NE) in the cued and the

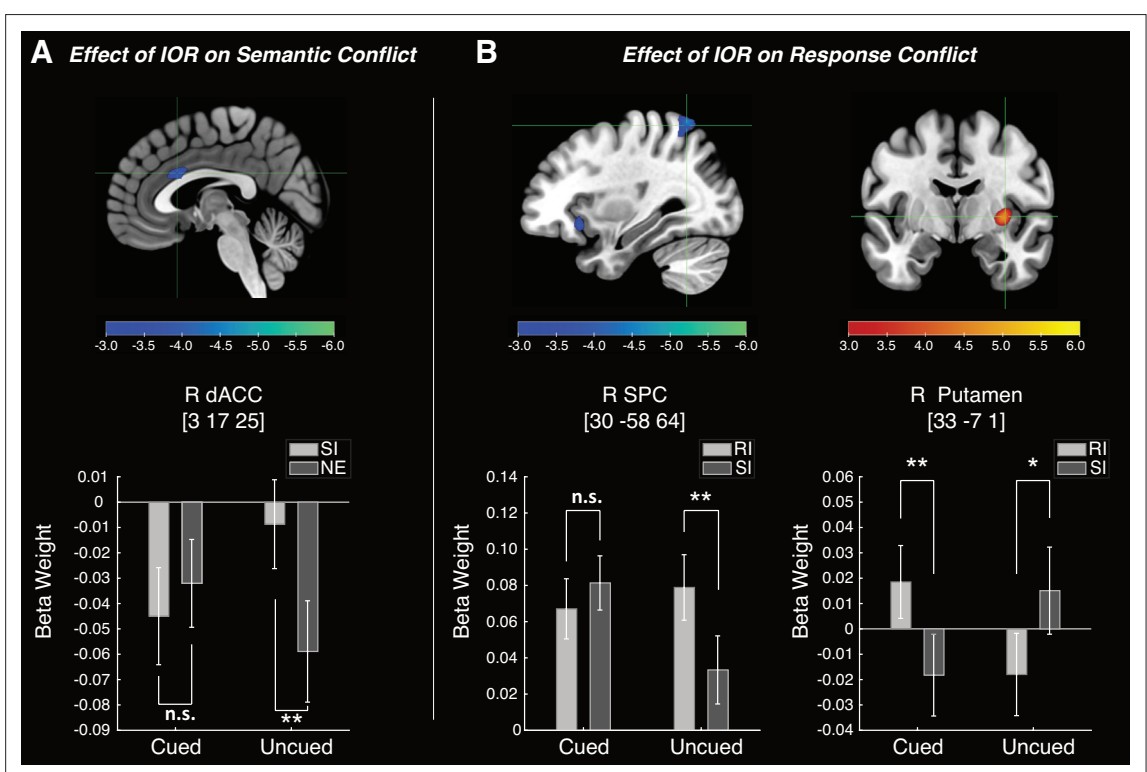

**Figure 4.** Effect of inhibition of return (IOR) in semantic conflict and response conflict. (**A**) Regions showing the IOR modulation of semantic conflict, defined as (cued-SI – cued-NE) > (uncued-SI – uncued NE). (**B**) Regions showing the IOR modulation of response conflict, defined as (cued-RI – cued-SI) > (uncued-RI – uncued-SI). NE = neutral; SI = semantically incongruent; RI = response-incongruent. Parameter estimations were based on a threshold of p<0.005 (uncorrected), with a minimum cluster size of 540 mm³ (20 voxels), yielding a threshold of corrected p<0.05 based on 2500 Monte Carlo simulations in BrainVoyager. Error bars extend to 1 SEM (N = 29). **p<0.01, *p<0.05, n.s.=nonsignificant.

**Table 2.** Brain regions showing a significant modulation effect of inhibition of return (IOR) on semantic conflict (cued-SI minus cued-NE >uncued SI minus uncued-NE) or response conflict (cued-RI minus cued-SI >uncued RI minus uncued-SI).

| Region | Laterality | Cluster (voxels) | MNI coordinates peak | | | $T_{max}$ | $T_{mean}$ | BA |
|---|---|---|---|---|---|---|---|---|
| | | | x | y | z | | | |
| Effect of IOR on semantic conflict | | | | | | | | |
| Dorsal anterior cingulate cortex | R | 24 | 3 | 17 | 25 | –3.78 | –3.34 | 24 |
| Effect of IOR on response conflict | | | | | | | | |
| Superior parietal cortex | R | 67 | 30 | –58 | 64 | –4.74 | –3.49 | 7 |
| Putamen | R | 31 | 33 | –7 | 1 | 4.91 | 3.80 | / |

MNI = Montreal Neurological Institute; BA = Brodmann area; R = right. '/' indicates that no BA could be assigned.

uncued conditions. As illustrated in **Figure 4A**, the right dACC showed significantly reduced activation. A two-way rm-ANOVA was conducted on the average parameter estimates (beta weights) obtained from these contrasts for each activated region. The results confirmed a significant interplay between semantic conflict and IOR in the right dACC, $F(1, 28) = 15.946$, $p < 0.001$, $\eta_p^2 = 0.363$. Greater neural activities were found in the SI condition compared to the NE condition when the targets were presented at the uncued location ($t(28) = 3.262$, $p = 0.003$, Cohen's $d = 0.606$), but not for the targets at the cued location ($t(28) = –1.010$, $p = 0.321$, Cohen's $d = 0.187$).

### Effect of IOR on response conflict

To explore the influence of IOR on response conflict, we compared the cued (RI-SI) and the uncued (RI-SI) conditions (**Figure 4B** and **Table 2**). The right superior parietal cortex (SPC) showed a significant activation reduction (**Figure 4B**, left), while the right putamen exhibited an activation enhancement (**Figure 4B**, right). A two-way rm-ANOVA on the beta weights revealed a significant interaction in the right SPC, $F(1, 28) = 20.833$, $p < 0.001$, $\eta_p^2 = 0.427$. Specifically, it showed greater activations in the RI condition compared to the SI condition when the targets were presented at the uncued location ($t(28) = 3.447$, $p = 0.002$, Cohen's $d = 0.640$), but not for the cued location ($t(28) = –0.962$, $p = 0.344$, Cohen's $d = 0.179$). The right putamen also demonstrated a significant interaction ($F(1, 28) = 26.686$, $p < 0.001$, $\eta_p^2 = 0.488$), but with a different pattern. The activation was stronger in the RI than the SI conditions for the cued location ($t(28) = 2.983$, $p = 0.006$, Cohen's $d = 0.554$); and the opposite pattern was observed for the uncued location ($t(28) = –2.404$, $p = 0.023$, Cohen's $d = 0.446$).

## Discussion
### Neural substrates of integration and segregation

The integration-segregation theory has emerged as an influential framework for explaining the dynamic effects of exogenous attention (**Chen et al., 2007**; **Funes et al., 2008**; **Hu et al., 2011**; **Li et al., 2018**; **Lupiáñez et al., 2001**; **Lupiáñez et al., 2007**; **Zu et al., 2023**), attributing the turning from the early attentional facilitation to the later IOR to the dynamic processes of cue-target integration and segregation (**Funes et al., 2008**; **Lupiáñez and Jesús Funes, 2005**; **Lupiáñez et al., 2001**; **Milliken et al., 2000**). In the current study, by contrasting the cued vs. uncued targets under long SOA, we provided the first direct neuroimaging evidence supporting this theory by dissociating brain activation patterns associated with these two processes. Specifically, the cued and uncued targets engaged distinct neural systems that respectively map onto the functional demands of re-engaging an existing representation and those of encoding a novel spatial event.

The heightened activations of the bilateral FEF and IPS, and right TPJ for targets appearing at the cued locations reflected the increased attentional demand associated with the integration process. Within the integration-segregation theoretical framework, this demand is conceptualized as the need to maintain or 're-open' an object file. Specifically, this framework posits that in the long SOA conditions, the cue-initiated object file is likely to have closed or begun closing, hindering

immediate integration of the subsequent targets (*Funes et al., 2008*; *Lupiáñez and Jesús Funes, 2005*; *Lupiáñez et al., 2001*; *Milliken et al., 2000*). Thus, to integrate a target appearing again at the cued location, the object file needs to be reopened with reallocation of attentional resources. Our neuroimaging data captured this process by showing coordinated activation in the bilateral FEF and IPS (key nodes of the DAN) and in the right TPJ (a core region of the VAN) (*Ahrens et al., 2019*; *Corbetta and Shulman, 2002*; *Fox et al., 2006*; *Vossel et al., 2014*). These regions act in concert to support the attentional shifts and reorienting necessary for reopening the object file for integration. Furthermore, the observed increase in the left dACC activity under the cued relative to the uncued condition likely reflected the engagement of cognitive control mechanisms (*Botvinick et al., 2004*; *Chung et al., 2024*; *Mayer et al., 2012*; *van Veen and Carter, 2005*), particularly in resolving the conflict between the task-driven requirement of target integration and the reduced accessibility of the cue-initiated representation. In this context, the heightened activation of dACC may also reflect its role in fulfilling the inhibitory bias toward the cued location (*Mayer et al., 2004b*) and discouraging inefficient integration attempts at a location marked as less relevant.

On the other hand, the recruitment of the bilateral PHG and the STG in the uncued condition supports the conceptualization of segregation as an active process of creating a new object file. According to the integration-segregation theory, when a target appears at an uncued location, the brain will register it as a new and separate event. Our results suggest that the brain recruits mechanisms specialized for spatial novelty to support this segregation process. Specifically, PHG is involved in episodic encoding of novel visual or spatial stimuli (*Burgess et al., 2002*; *Danieli et al., 2023*; *Hayes et al., 2007*; *Li et al., 2016*; *Menon et al., 2000*; *Ranganath and Rainer, 2003*; *Torres-Morales and Cansino, 2024*), while the STG supports the detection of salient and unexpected changes in the sensory environment (*Corbetta and Shulman, 2002*; *Downar et al., 2002*). Thus, the enhanced PHG and STG activities observed in the uncued condition may signify the active engagement of a novelty-detection system required for encoding new spatial representations and establishing new object files.

## Connections and discrepancies with previous IOR neuroimaging studies

Our data provided clear support for the integration-segregation theory. It is also noteworthy that, although prior studies investigated the neural mechanisms of IOR (*Bourgeois et al., 2013a*; *Bourgeois et al., 2013b*; *Hanlon et al., 2017*; *Lepsien and Pollmann, 2002*; *Mayer et al., 2004a*; *Mayer et al., 2007*; *Mayer et al., 2004b*; *Müller and Kleinschmidt, 2007*; *Satel et al., 2019*; *Yang and Mayer, 2014*; *Zhou and Chen, 2008*), none identified distinct activation patterns corresponding to the integration and segregation processes as in our data. Specifically, most of the previous IOR studies did not show significant brain activations when contrasting the cued and uncued conditions (*Lepsien and Pollmann, 2002*; *Mayer et al., 2004b*), except that *Chen et al., 2006*, reported a cue-validity effect confined to the left FEF. Instead, some indirect approaches, such as comparing long- and short-SOA trials while collapsing over the cueing conditions, reported activations in regions like the FEF, TPJ, ACC, and posterior parietal cortex (*Lepsien and Pollmann, 2002*; *Mayer et al., 2004a*; *Mayer et al., 2004b*; *Müller and Kleinschmidt, 2007*; *Zhou and Chen, 2008*), showing some similarity with the integration-related network observed in the current study. However, the findings were inconsistent across studies, with some reporting only a limited subset of regions and others showing lateralized instead of bilateral effects (e.g. stronger right-hemisphere FEF activation; *Lepsien and Pollmann, 2002*; *Mayer et al., 2004a*). Similar frontoparietal engagement has also been observed in auditory and cross-modal IOR studies (*Hanlon et al., 2017*; *Mayer et al., 2009*; *Mayer et al., 2007*; *Yang and Mayer, 2014*), typically present across various SOAs (e.g. sustained activation in both the dorsal and ventral frontoparietal regions regardless of SOA length; *Hanlon et al., 2017*) or showing SOA-dependent effects (e.g. reversed direction of activation differences between short and long SOAs; *Mayer et al., 2007*). Complementing these observations, transcranial magnetic stimulation (TMS) studies have provided causal evidence for the contribution of frontoparietal regions in IOR (*Bourgeois et al., 2013a*; *Bourgeois et al., 2013b*; *Chica et al., 2011*; *Ro et al., 2003*). For instance, stimulation over the right FEF during the cue-target interval has been shown to eliminate the typical IOR effect for the cued targets in the ipsilateral hemifield (*Ro et al., 2003*). Similarly, TMS applied to the right IPS/TPJ also disrupted the IOR effect (*Bourgeois et al., 2013a*; *Chica et al., 2011*), whereas stimulation over their left-hemisphere counterparts did not cause much change in IOR (*Bourgeois et al., 2013b*). These findings suggest a possible right-lateralized neural organization of the

integration process. However, this lateralization notion conflicts with the largely bilateral activation pattern observed in our study. The lack of systematic testing for the left-hemisphere contribution in previous TMS studies leaves this asymmetry open to further investigation. Notably, despite offering partial (and often lateralized) support for the integration process, none of these prior studies have addressed the neural mechanisms underlying the segregation process, which is uniquely revealed by the present neuroimaging findings.

The above discrepancies between our findings and the previous studies may stem from several methodological and design factors. Firstly, the prior studies likely introduced confounds when investigating IOR indirectly. When comparing long and short SOAs, the observed effects may have been jointly influenced by factors unrelated to IOR, such as working memory (i.e. increased demand of maintaining the cue representation over longer intervals; *Mayer et al., 2007*) and temporal attention (i.e. distinct temporal expectations formed by variations in SOA; *Nobre and van Ede, 2018*). Moreover, the IOR effect depends not only on cue-induced attentional orienting, but also on the dynamic interaction between the target onset and the ongoing cue-related neural activity (*Lupiáñez, 2010*; *Nobre and van Ede, 2018*; *Taylor and Donnelly, 2002*). These confounds could potentially obscure the genuine IOR effect. Second, differences in statistical power may also account for the discrepancies. In the present study, we employed an optimized GA stimulus sequence (*Wager and Nichols, 2003*), which provides greater statistical power than simple random sequences while maintaining a high estimation efficiency (for details, see the Methods and Supplementary Information sections). This optimization likely enhanced the reliability of the estimated neural responses (*Wager and Nichols, 2003*). In addition, the previous neuroimaging studies on IOR often relied on relatively small sample sizes (around 10–12 participants; *Chen et al., 2006*; *Mayer et al., 2004a*; *Mayer et al., 2004b*; *Müller and Kleinschmidt, 2007*) or a limited number of trials (e.g. 30 trials per condition; *Lepsien and Pollmann, 2002*), leading to much reduced statistical power and a higher probability of false negatives. In contrast, the current study increased both the number of trials and the sample size, effectively enhancing the sensitivity of detecting differences between experimental conditions (*Baker et al., 2021*; *Chen et al., 2022*). Finally, task design differences may further contribute to the observed inconsistencies. The earlier studies often employed simple localization or detection tasks (*Lepsien and Pollmann, 2002*; *Mayer et al., 2004a*; *Mayer et al., 2004b*; *Müller and Kleinschmidt, 2007*), while the current study adopted a discrimination task. According to the integration-segregation theory (*Funes et al., 2008*; *Lupiáñez and Jesús Funes, 2005*; *Lupiáñez et al., 2001*; *Lupiáñez et al., 2007*; *Milliken et al., 2000*), more complex stimuli may require greater cognitive resources to establish object files, leading to enhanced processing of object files and heightened detectability of the underlying integration and segregation processes.

## Neural interactions between IOR and cognitive conflict processing

Another novelty of the current study is integrating the IOR and the modified Stroop tasks, which were separately studied for the semantic- and response-related conflicts (*De Houwer, 2003*; *van Veen and Carter, 2005*). Through this design, we made an additional discovery about how IOR modulates the ongoing Stroop interference effect at the inhibited (i.e. cued) locations. Behaviorally, our results showed no significant interaction between IOR and any conflict in the Stroop task, not replicating the previous findings (*Chen et al., 2006*; *Vivas and Fuentes, 2001*) of reduced Stroop interference at the cued relative to the uncued locations. Yet, at the neural level, the brain regions involved in conflict processing were engaged in the interaction between IOR and the Stroop effect. Specifically, the right dACC, which is involved in semantic conflict processing (*Li et al., 2017*; *Milham et al., 2001*; *van Veen and Carter, 2005*), appeared to serve as a critical neural interface for the interaction between semantic conflict and IOR. Specifically, in the uncued condition, the semantic incongruency elicited stronger activations compared to the neutral condition, a pattern that disappeared or even reversed in the cued condition. Regarding the interaction between response conflict and IOR, brain regions such as the right SPC, which are involved in detecting response conflict and orienting spatial attention (*Li et al., 2017*), played a key role. Similarly, this region exhibited stronger conflict effects (i.e. greater activation in the RI than SI condition) in the uncued condition compared to the cued condition. These results can be interpreted by the inhibitory tagging mechanism proposed by *Fuentes et al., 1999*, which posits that, when attention is drawn away from a cued location, stimuli presented there are temporarily tagged with inhibition (*Fuentes et al., 2000*; *Fuentes et al., 1999*; *Vivas and Fuentes,*

*2001*). ERP evidence supporting this mechanism was reported by *Zhang et al., 2012*, who showed that the Stroop conflict-related N450 effects were delayed and attenuated at the cued compared to the uncued locations, suggesting a temporary disruption of the stimulus-response link. Such inhibitory tagging may attenuate or even disrupt conflict processing at the inhibited location, offering a plausible account for the neural interactions between IOR and Stroop conflicts observed in our study. The current results could also potentially suggest that the effects of inhibitory tagging are not limited to stimulus-response connections (as proposed by *Fuentes et al., 1999*), but also extend to semantic representations, as evidenced by the modulation of the right dACC observed in our study. This notion is consistent with a previous finding that the N400 ERP component (a biomarker of semantic processing) had a decreased amplitude for the cued position (*Zhang and Zhang, 2007*). This highlights that spatial attention can affect subsequent cognitive processes at the semantic level (*Cristescu and Nobre, 2008*; *Zhang and Zhang, 2007*).

Furthermore, we observed pronounced neural responses in the right putamen when contrasting the RI and SI conditions at the cued vs. the uncued locations. The putamen is a subcortical nucleus in the basal ganglia and has been found to be involved in the control of response interference (*Schmidt et al., 2018*; *Schmidt et al., 2020*). For example, *Schmidt et al., 2020*, demonstrated that the dorsal striatum, including the putamen, is engaged during Simon-type interference by supporting task-appropriate response selection and suppression of competing alternatives, and that its damage leads to less efficient interference control (*Schmidt et al., 2018*). These findings support the view that the putamen is recruited when interference arises at the response-selection level. Building on this, we speculated that the enhanced putamen activation in the cued conditions in the current study reflects an increased demand for response control when attentional resources were reduced by IOR. Taken together, our findings highlight a potential neural basis for the interaction between IOR and conflict processing encompassing both semantic and response domains.

## Methodological considerations and limitations

While the neural interactions between IOR and conflict processing offer novel insights, we need to be cautious when interpreting the results given that the neural interactions were observed without any corresponding behavioral effect. One likely explanation for this dissociation is the differences in measurement sensitivity between the behavioral and neural indices (*Chen et al., 2006*; *Wilkinson and Halligan, 2004*). As noted by *Wilkinson and Halligan, 2004*, RTs and accuracies are not perfect measures of cognition, whereas neural signals can reveal finer-grained or 'hidden' processes that precede overt behavior. Consistent with this view, *Chen et al., 2006*, reported a similar dissociation in which response conflict modulation by IOR was clearly seen in neural data but not in behavior. This suggests that neural modulations sometimes emerge even in the absence of detectable behavioral differences. In addition, the usage of a GA-optimized sequence in the current study may have partly accounted for the observed dissociation. While this optimization enhances both detection and HRF estimation efficiency, it may result in partially clustered event sequences that resemble a block-like structure, thereby reducing event counterbalancing and increasing sequential predictability (*Wager and Nichols, 2003*). As a result, the participants may have formed expectations about upcoming events and weakened the correspondence between the neural and behavioral findings. Future studies are required to address this limitation by employing more optimized designs that consider some psychological factors (e.g. event counterbalancing; *Wager and Nichols, 2003*) to better validate the observed neural mechanisms.

## Conclusion

In conclusion, the current study provides the first direct neuroimaging evidence lending support to the hypothesis of the integration-segregation theory (*Funes et al., 2008*; *Lupiáñez and Jesús Funes, 2005*; *Lupiáñez et al., 2001*; *Milliken et al., 2000*). We revealed distinct neural mechanisms for processing of the cued and uncued targets during IOR, with attentional integration engaging the frontoparietal attention network (FEF, IPS, TPJ, dACC) and segregation recruiting the medial temporal regions (PHG-STG) associated with new object-file formation and novelty encoding. These dissociated activations offered direct support for the dynamic interplay between the integration and segregation processes. We also identified interactions between IOR and cognitive conflict in brain activities, suggesting that attentional orienting can modulate conflict processing at both the semantic and

response levels. Taken together, our findings revealed the neural underpinnings of the integration-segregation theory and advanced our understanding of the neural mechanisms linking exogenous attentional orienting and cognitive control.

## Methods

### Participants

32 healthy participants with normal or corrected-to-normal vision and normal color vision were recruited. All participants were right-handed and reported no history of neurological or psychiatric disorders. Data from three participants were excluded due to excessive head movements and high global variances (see fMRI data analysis), leaving 29 participants for analysis (18 female, 11 male; aged 18–30 years, M=22.69, SD = 2.58). All participants were naïve to the purpose of the study, provided written informed consent approved by the Ethics Committees of Northeast Normal University and Soochow University (SUDA20240119H01), and received monetary compensation. The sample size was informed by a power analysis using MorePower 6.0 (*Campbell and Thompson, 2012*) for a within-subjects rm-ANOVA. To achieve an 80% statistical power at the threshold of $\alpha$=0.05 (*Chen et al., 2006*), 14 participants were required. In addition, we also acknowledged that effect sizes from published studies are often inflated due to the publication bias (*Albers and Lakens, 2018*). To mitigate this potential risk, we determined to acquire data from a sample at least twice the size suggested by the power analysis (i.e. $N{\geq}28$).

### Experimental design

The experiment adopted a within-subjects design with two factors, namely cue validity (cued and uncued) and congruency (SI, RI, and NE). The targets appeared at the cued location on the cued trials and at the other peripheral location on the uncued trials. The congruency factor referred to the relationship between the ink color and the meaning of the Chinese characters (i.e. targets) according to the predefined stimulus-response mapping. In total, eight characters and four colors were used (see *Figure 1B*). Ink colors of the stimuli were mapped onto two response keys, with red and green assigned to one response key (Key 1) while yellow and blue assigned to the other response key (Key 2). In an SI trial, the ink color and the character meaning were incongruent but mapped to the same response key (e.g. character '红' [meaning 'red'] being displayed in green ink, with both colors being mapped to the same response key). Thus, the SI trials involved a semantic conflict without inducing a response conflict. In an RI trial, the ink color and the character meaning differed and were also mapped to different response keys (e.g. '红' [red] being displayed in yellow ink, with red and yellow being related to different responses), leading to both the semantic and response conflicts. The NE trials used characters that were not related to any color in their meanings and shared the same orthographic structures (character complexity and form) as the color characters. In addition to the six experimental conditions, a null condition (no Chinese character presented) was included as an implicit baseline to facilitate estimation of the effects of interest in the ER-fMRI analysis (*Burock et al., 1998*; *Friston et al., 1999*; *Liu, 2004*).

It is worth noting that the statistical power of effects in rapid ER-fMRI depends greatly on specific sequences of stimulus events (*Liu and Frank, 2004*; *Wager and Nichols, 2003*). To ensure high design efficiency, we optimized the stimulus sequences employing the GA (see Supplementary Information for details) (*Wager and Nichols, 2003*). This optimization improves the detection efficiency for the contrasts of interest by moderately sacrificing the efficiency of less relevant contrasts (*Wager and Nichols, 2003*). In the current study, we focused on three contrasts, including cued-NE vs. uncued-NE, cued-SI minus cued-NE vs. uncued-SI minus uncued-NE, and cued-RI minus cued-SI vs. uncued-RI minus uncued-SI. These contrasts respectively examined the IOR effect, the modulation of semantic conflict processing by IOR, and the modulation of response conflict processing by IOR. The optimized sequences were used for all but two participants, whose trial sequences were constructed using a truncated M-sequence (*Buracas and Boynton, 2002*) implemented in an earlier version of the experiment.

### Stimuli and procedure

Each participant completed two functional scans (i.e. experimental runs) and one anatomical scan in a single session. Each experimental run employed a rapid event-related design and had each of the

seven conditions (six experimental conditions plus the null condition) repeated 48 times (336 trials per run). Across the two runs, this yielded a total of 672 trials (96 trials per condition).

All trials displayed a three-box display over a gray background, including a central black fixation box (1°×1°, line width of 0.02°) and two black placeholder boxes (1.5°×1.5°, line width of 0.02°) positioned 4° (center-to-center) to the left and right of the fixation box. Each run began and ended with this display for 16 and 20 s, respectively. The trial sequence is illustrated in *Figure 1A*. In a null trial, only the three boxes were shown for the trial duration. In any of the six experimental conditions, each trial started with one of the peripheral boxes changing to a white color with a line width of 0.05° for 150 ms to attract attention to this peripheral location (cue). 150 ms after the offset of the peripheral cue, the central fixation box turned into white with a line width of 0.05° for 150 ms to force attention back to the central location (central cue). After another 450 ms, a colored Chinese character (in the STSong font, 1.4°×1.4°) was presented (target) for 450 ms inside one of the two peripheral boxes with equal probabilities. Participants were required to ignore the meaning of the character and identify the word color as quickly and accurately as possible by pressing one of the two keys designated for the color categories (red/green and blue/yellow) with their middle and index fingers, respectively (*Figure 1B*). The color category-button mapping was counterbalanced across participants. Furthermore, to avoid a possible occurrence of the Simon effect (*Klein and Ivanoff, 2011*), the response keys were vertically arranged. Each trial ended with an intertrial interval with a duration of 850, 1050, 1250, or 1450 ms (randomized with equal probabilities). The average trial duration was 2500 ms.

Before the scanning, all participants had two practice parts outside the scanner to familiarize themselves with the task and the stimuli. In the first part, the participants practiced on a discrimination task with only color patches (no Chinese characters) using the predefined color category-button mapping. Once having reached an accuracy of 96%, the participants did the second part and completed 24 practice trials of the experimental task, as in the scanning runs.

## Apparatus and data acquisition

The imaging data were acquired at two research sites following comparable protocols, with equal numbers of participants scanned at each site (*N*=16 per site). At the Imaging Center for Brain Research of Beijing Normal University, the stimuli were presented with E-Prime (Psychological Software Tools, Pittsburgh, PA, USA) on an LCD monitor (1024×768 resolution; 60 Hz refresh rate; see *Zhang et al., 2018*, for spatiotemporal properties) viewed through a head-coil-mounted mirror at an optical distance of 115 cm. The data were collected using a Siemens 3-Tesla Tim Trio scanner with a head coil. The functional data were acquired through a T2*-weighted echo planar imaging (EPI) sequence (TR = 2000 ms; TE = 30 ms; flip angle = 90°; FOV = 220 × 220 mm$^2$; matrix size = 64 × 64). Thirty-three transversal slices covering the whole brain (slice thickness = 4 mm; in-plane resolution = 3.44 × 3.44 mm$^2$; slice gap = 0.4 mm) were acquired in an interleaved ascending order. Each participant completed two functional runs of 400 volumes (including 8 initial dummy volumes). High-resolution anatomic images were collected using a T1*-weighted magnetization-prepared rapid gradient echo (MP-RAGE) sequence consisting of 128 sagittal slices (TR = 2300 ms; TE = 3.9 ms; flip angle = 8°; FOV = 256 × 256 mm$^2$, matrix size = 256 × 256, voxel resolution = 1.33× 1×1 mm$^3$, slice gap = 0 mm). Responses were collected using an MRI-compatible 2-button fiber-optic response pad (*Tang et al., 2025*).

At the Imaging Center of the First Affiliated Hospital of Soochow University, the stimuli were presented with MATLAB (The MathWorks, Natick, MA, USA) and the Psychophysics Toolbox (*Brainard, 1997*) on an LCD monitor (1920×1080 and 60 Hz) viewed through a mirror at an optical distance of 251 cm. The imaging data were recorded using a 3-Tesla Philips Ingenia scanner equipped with a head coil. The functional images featured a matrix size of 80×80, an in-plane resolution of 2.75×2.74 mm$^2$, and no slice gap. The structural images were acquired with a voxel resolution of 1×1×1 mm$^3$ across 180 slices (FOV = 240 × 240 mm$^2$; matrix size = 240 × 240). The other parameters remained the same as those used at Beijing Normal University. Responses were collected using an MRI-compatible 2-button fiber-optic response pad.

## Data analysis

### Behavioral analysis

Trials with incorrect responses (4.29% of trials) and RT outliers (1.24% trials, RTs shorter than 150 ms, considered anticipatory responses, or longer than 1300 ms, reflecting extremely slow responses) were excluded from statistical analyses (*Ratcliff, 1993*; *Whelan, 2008*). Mean RTs on correct trials and response accuracies were entered into the two-way rm-ANOVA.

### fMRI data analysis

The fMRI preprocessing and analysis were conducted with the BrainVoyager QX (version 2.2, Brain Innovation) software package (*Goebel et al., 2006*). The initial eight functional volumes of each scan were discarded to allow signal equilibration. For the remaining functional images, slice timing correction was applied using sinc interpolation, followed by 3D motion correction with trilinear/sinc interpolation for intra-session alignment to the middle volume. Each run for each participant was examined for the six head motion parameters (three rotations and three translations). Runs with motions exceeding one voxel length in any direction were excluded (resulting in the exclusion of two runs). An isotropic Gaussian kernel of an 8 mm full width at half maximum was then applied to spatially smooth the images. Finally, linear trend removal was performed, along with high-pass temporal filtering at a cutoff of approximately 0.0081 Hz (corresponding to seven cycles per run), to remove low-frequency nonlinear drifts. Data quality was further assessed using the variance in the global signal (mean signal across all voxels within each run). Runs with global variance ≥0.1% were excluded, resulting in the exclusion of eight runs (see Supplementary Information for details). Ultimately, three participants were excluded because neither run met the quality criteria. All remaining participants retained both runs, except for three individuals who each contributed only one valid run. The retained functional images were then co-registered to each participant's high-resolution anatomical scan in native space and subsequently normalized to the Montreal Neurological Institute (MNI) 152 template, with a resampled voxel size of $3 \times 3 \times 3$ mm$^3$.

After the preprocessing, statistical analyses were performed using a random effects general linear model (RFX-GLM) analysis within BrainVoyager, executing a multi-subject GLM with distinct predictors for each participant. Using a deconvolution and multiple regression approach, we modeled six experimental conditions and one 'error' term (including all the error trials) for each participant, with each condition including six sampling points taken from the 0–12 s period after the cue presentation (i.e. one sampling point every 2 s). The functional images occurring 5–10 s after the cue onsets, corresponding to the peak of the HRF (*Cohen et al., 1997*), were used to provide parameter estimates for the amplitudes of the HRF. Volumes deviating in intensity by ± 3 SDs or more from the individual means were removed by a weighted vector that was included in the model as a covariate of no interest. In addition, the six mean-centered head motion parameters were modeled as covariates of no interest to further remove any residual variance due to head motion. To mitigate noise related to global physiological processes, the model incorporated the global signal, which represented the normalized average activity across all voxels at each time point in the standard space, as an additional predictor. We examined the three contrasts of interest introduced earlier. Corrections for multiple comparisons at p<0.05 were made through the Cluster Threshold plugin (BrainVoyager) using 2500 Monte Carlo simulations. Minimum cluster sizes (540 mm$^3$ corresponding to 20 voxels) corresponding to significance at a threshold of p<0.005 (uncorrected) were computed for each contrast (*Forman et al., 1995*). The approximate Brodmann areas (BAs) and the corresponding anatomical labels of the peak voxel of the significant clusters in the MNI space were identified using the Neuroelf toolbox v1.1 (*Weber, 2017*).

## Acknowledgements

This research was supported by grants from the Brain Science and Brain-like Intelligence Technology-National Science and Technology Major Project (2025ZD0215702), the National Natural Science Foundation of China (32171049), the Social Science Foundation of Jiangsu Province (22JYB015), the China Postdoctoral Science Foundation (2024M752310), Jiangsu Funding Program for Excellent Postdoctoral Talent (2024ZB496, 2025ZB642), and Natural Science Foundation of Jiangsu Province (BK20250796).

# Additional information

## Funding

| Funder | Grant reference number | Author |
| --- | --- | --- |
| Brain Science and Brain-like Intelligence Technology - National Science and Technology Major Project | 2025ZD0215702 | Yang Zhang |
| National Natural Science Foundation of China | 32171049 | Yang Zhang |
| Social Science Foundation of Jiangsu Province | 22JYB015 | Yang Zhang |
| Postdoctoral Research Foundation of China | 2024M752310 | Yujie Chen |
| Jiangsu Funding Program for Excellent Postdoctoral Talent | 2024ZB496 | Yujie Chen |
| Jiangsu Funding Program for Excellent Postdoctoral Talent | 2025ZB642 | Ai-Su Li |
| Natural Science Foundation of Jiangsu Province | BK20250796 | Ai-Su Li |

The funders had no role in study design, data collection and interpretation, or the decision to submit the work for publication.

## Author contributions

Yujie Chen, Data curation, Formal analysis, Validation, Visualization, Methodology, Writing – original draft, Writing – review and editing; Ai-Su Li, Investigation, Visualization, Writing – original draft, Writing – review and editing; Yang Yu, Su Hu, Resources, Data curation; Xun He, Writing – original draft, Writing – review and editing; Yang Zhang, Conceptualization, Resources, Data curation, Software, Formal analysis, Supervision, Funding acquisition, Investigation, Visualization, Methodology, Writing – original draft, Project administration, Writing – review and editing

## Author ORCIDs

Yujie Chen (iD) https://orcid.org/0000-0001-5910-382X
Ai-Su Li (iD) https://orcid.org/0000-0002-2340-3422
Yang Zhang (iD) https://orcid.org/0000-0002-1827-9383

## Ethics

All participants were naïve to the purpose of the study, provided written informed consent approved by the Ethics Committees of Northeast Normal University and Soochow University (SUDA20240119H01), and received monetary compensation.

Reviewer #1 (Public review): https://doi.org/10.7554/eLife.109842.3.sa1
Reviewer #2 (Public review): https://doi.org/10.7554/eLife.109842.3.sa2
Reviewer #3 (Public review): https://doi.org/10.7554/eLife.109842.3.sa3
Author response https://doi.org/10.7554/eLife.109842.3.sa4

# Additional files

## Supplementary files

MDAR checklist

## Data availability

The datasets and codes for this study are available at Open Science Framework https://osf.io/av4xn or GitHub https://github.com/yangzhangpsy/ER-fMRI-IOR (copy archived at *Zhang, 2025*).

The following dataset was generated:

| Author(s) | Year | Dataset title | Dataset URL | Database and Identifier |
|---|---|---|---|---|
| Chen Y | 2026 | Data from: Dissociable neural substrates of integration and segregation in exogenous attention | https://osf.io/av4xn | Open Science Framework, av4xn |

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

## Appendix 1

### Sequence optimization

The statistical power of effects in rapid ER-fMRI depends greatly on the particular sequence of events chosen (*Liu and Frank, 2004*; *Wager and Nichols, 2003*). Here, we compared the performance of GA (*Wager and Nichols, 2003*), M-sequence (*Liu, 2004*; *Liu and Frank, 2004*; *Liu et al., 2001*), and random sequences under two different settings of ISI (*Burock et al., 1998*; *Dale, 1999*). Using simulation, we evaluated how these different stimulus sequences contribute to statistical power in the current study, with a specific focus on detection efficiency (i.e. the sensitivity of detecting activations) and estimation efficiency (i.e. the ability to accurately estimate the HRF).

Specifically, following the definitions by *Wager and Nichols, 2003*, the efficiency of a sequence is defined as:

$$\xi = 1/trace\left\{ diag(w)\, CZ^{-}KVK'\left(Z^{-}\right)' C' \right\} \tag{1}$$

where $diag(w)$ is a diagonal matrix comprised of the elements of $w$ (the weight vector for the prior comparisons); $Z$ represents the filtered design matrix ($Z=KX$, $K$: filtering matrix; $X$: design matrix); and $V$ is the correlation matrix of the errors. When $C$ is the identity matrix, $\xi$ reflects the estimation efficiency for the HRF shape. Alternatively, when $C$ represents the canonical HRF, it corresponds to the detection efficiency for the prior contrasts.

Additionally, we incorporated a high-pass filter with a cutoff time constant of 128 s. To mitigate any nonlinear effects from ISIs shorter than 2 s, we set the activation upper bound to twice the maximum value of the classic HRF function (*Wager and Nichols, 2003*). Moreover, we used the first-order autoregressive model (*Friston et al., 2000*) to account for autocorrelation. Using the formula and the settings described above, the detection efficiency and estimation efficiency can be calculated for any stimulus sequence.

The simulation procedure was as follows: (1) using the GA toolbox provided by *Wager and Nichols, 2003*, we generated multiple GA stimulus sequences tailored to this experiment (300 runs, 30 repetitions each); (2) M-sequences were generated using the Mseq2 program (*Liu, 2004*; *Liu and Frank, 2004*); (3) 500 random sequences were created using a custom Optseq2 algorithm, and the best-performing ones were selected; (4) these three optimal sequences underwent ISI jittering (2.2 s, 2.4 s, 2.6 s, 2.8 s) for 500 iterations.

*Appendix 1—figure 1* presents the detection efficiency (A) and estimation efficiency (B) for the different stimulus sequences. The simulation results indicate that, in the fixed (i.e. non-jitter) ISI condition, the GA-generated sequence exhibited the highest detection efficiency (approximately 30), followed by the random sequence (max value around 25.5), then the M-sequence, which had the lowest detection efficiency (about 20). After jittering the ISI, the trend in the detection efficiency remained consistent. In terms of the HRF estimation efficiency, the M-sequence performed best in both the jittered and fixed ISI conditions, followed by the GA sequence, while the random sequence showed the lowest efficiency. Notably, ISI jittering significantly improved the HRF estimation efficiency, with all three sequences performing better under jittered ISI compared to fixed ISI.

In summary, the use of the GA sequences effectively enhanced the detection efficiency. Moreover, by implementing a jittered ISI design, the algorithm ensured the optimal HRF estimation efficiency.

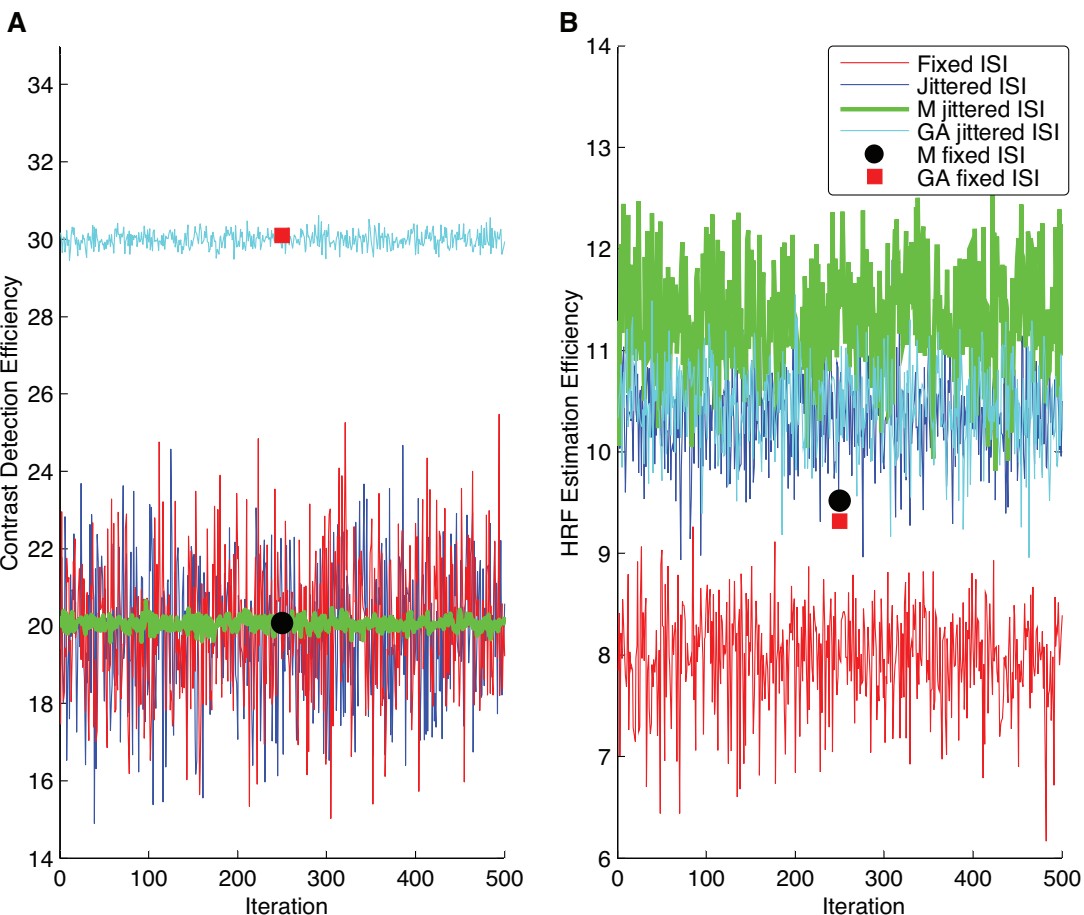

**Appendix 1—figure 1.** Simulation results of different stimulus sequences. (**A**) Contrast detection efficiency for the effects of interest. (**B**) Hemodynamic response function (HRF) estimation efficiency for the hemodynamic response shape.

## Global variance quality control

Global variance was used as a quality-control index to identify runs with abnormal signal variance. This procedure was applied after the standard preprocessing steps (including slice timing correction, 3D motion correction, spatial smoothing, linear trend removal, and high-pass temporal filtering). For each run, the global signal was computed as the mean signal across voxels with intensity values greater than 100 in each volume, thereby excluding background and non-brain voxels. The 100 threshold was applied to effectively distinguish brain tissue from low-intensity background noise. This resulted in a single global signal value per volume (TR). Global variance was then computed as the variance of the percentage global signal normalized to the mean (baseline). Specifically, for each run, the global signal time course $GS_t$ was first normalized to its run mean $\bar{GS}$ and expressed in percentage units as $x_t = 100 \cdot \frac{GS_t}{\bar{GS}}$. Global variance was then computed as the sample variance of $x_t$, defined as $\frac{1}{N-1} \sum_{t=1}^{N} (x_t - \bar{x})^2$, where $N$ denotes the number of time points. Runs with global variance values equal to or over 0.1% were considered demonstrating abnormal signal variances (i.e. signal instability) and therefore excluded from further analyses. *Appendix 1—figure 2* shows the global signal time series of all the excluded runs that exceeded the 0.1% global variance threshold. Each panel corresponds to one excluded run, with the global variance value indicated in the panel title. The figure shows the excluded runs from Subjects 12 and 16 scanned at the Imaging Center for Brain Research, Beijing Normal University, and Subjects 14, 21, and 22 scanned at the Imaging Center of the First Affiliated Hospital of Soochow University.

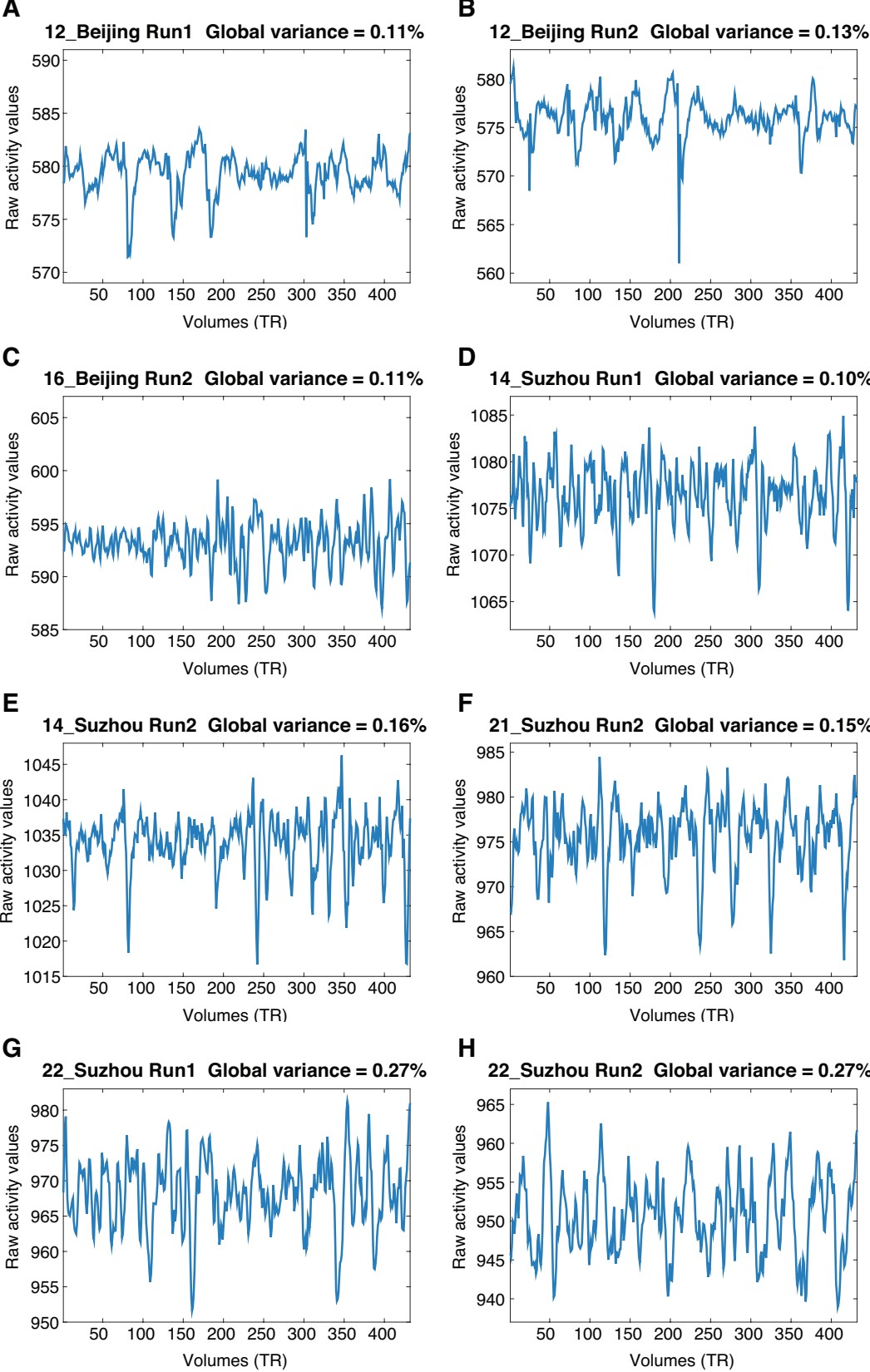

**Appendix 1—figure 2.** Global signal time series for runs excluded from further analysis. Each panel shows the global signal time course of one run that had a global variance over the exclusion threshold (≥0.1%). Eight runs
*Appendix 1—figure 2 continued on next page*

*Appendix 1—figure 2 continued*

were excluded across Subjects 12 and 16 (Imaging Center for Brain Research, Beijing Normal University) and Subjects 14, 21, and 22 (Imaging Center of the First Affiliated Hospital of Soochow University).

