## [Editor Report · eLife Assessment]

This **important** study uses an optimized IOR-Stroop fMRI paradigm to dissociate integration and segregation processes and to show that attentional orienting modulates conflict processing at both the semantic and response levels. The evidence is **compelling**, supporting the integration-segregation theory of exogenous attention in inhibition of return while also deepening our understanding of how attentional orienting shapes downstream cognitive processing. The work will therefore be of broad interest to researchers in attention and cognitive control.

---

## [Referee Report · Reviewer #1 (Public review)]

Summary:

This study makes a significant and timely contribution to the field of attention research. By providing the first direct neuroimaging evidence for the integration-segregation theory of exogenous attention, it fills a critical gap in our understanding of the neural mechanisms underlying inhibition of return (IOR). The authors employ a carefully optimized cue-target paradigm combined with fMRI to elegantly dissociate the neural substrates of cue-target integration from those of segregation, thereby offering compelling support for the integration-segregation account. Beyond validating a key theoretical hypothesis, the study also uncovers an interaction between spatial orienting and cognitive conflict processing, suggesting that exogenous attention modulate conflict processing at both semantic and response levels. This finding shed new light on the neural mechanisms that connect exogenous attentional orienting with cognitive control.

Strengths:

The experimental design is rigorous, the analyses are thorough, and the interpretation is well grounded in the literature. The manuscript is clearly written, logically structured, and addresses a theoretically important question. Overall, this is an excellent, high-impact study that advances both theoretical and neural models of attention.

Comments on revisions:

I appreciate the authors' thorough and thoughtful revisions, which have successfully addressed all of my prior concerns.

---

## [Referee Report · Reviewer #2 (Public review)]

This study provides neuroimaging evidence supporting the integration-segregation theory of inhibition of return (IOR), a widely studied attentional phenomenon. It also explores the neural interactions between IOR and cognitive conflict, demonstrating that conflict processing is potentially modulated by attentional orienting.

The integration-segregation theory was investigated using a sophisticated, well-executed experimental task that accounted for cognitive conflict processing, which is phenomenologically related to IOR but is non-spatial. The behavioral and neuroimaging data were carefully analyzed.

The authors have thoughtfully addressed all my previous concerns. By demonstrating how attentional orienting can modulate neural processing of cognitive conflict, this study helps to advance a more unified and mechanistic understanding of the cognitive and neural processes that govern our visual perception and response selection.

---

## [Referee Report · Reviewer #3 (Public review)]

Summary:

This study provides direct neuroimaging evidence relevant to the integration-segregation theory of exogenous attention-a framework that has shaped behavioral research for more than two decades but has lacked clear neural validation. By combining an inhibition-of-return (IOR) paradigm with a modified Stroop task in an optimized event-related fMRI design, the authors examine how attentional integration and segregation processes are implemented at the neural level and how these processes interact with semantic and response conflicts. The central goal is to map the distinct neural substrates associated with integration and segregation and to clarify how IOR influences conflict processing in the brain.

Strengths:

The study is well-motivated, addressing a theoretically important gap in the attention literature by directly testing a long-standing behavioral framework with neuroimaging methods. The experimental approach is creative: integrating IOR with a Stroop manipulation expands the theoretical relevance of the paradigm, and the use of a genetic-algorithm-optimized fMRI design ensures high efficiency. Methodologically, the study is rigorous, with appropriate preprocessing, modeling, and converging analyses across multiple contrasts. The results are theoretically coherent, demonstrating plausible dissociations between integration-related activity in the fronto-parietal attention network (e.g., FEF, IPS, TPJ, dACC) and segregation-related activity in medial temporal regions (e.g., PHG, STG). Importantly, the findings provide much-needed neural support for the integration-segregation framework and clarify how IOR modulates conflict processing.

Revisions and Evaluation:

The authors have responded thoroughly and convincingly to the concerns raised in the previous round of review. In particular, issues related to the interpretation of dACC activity, the functional characterization of PHG and STG, and reporting clarity have been carefully addressed. The manuscript has been improved in terms of transparency, consistency of reporting, and overall readability.

As a result, I no longer see any major weaknesses. The study is now clearly presented, methodologically sound, and theoretically informative. It makes a valuable contribution to the literature on attention and cognitive control.

Comments on revisions:

I appreciate the authors' efforts in addressing the previous comments. They have responded thoroughly to the concerns raised in the prior round of review. The work is well executed and makes a meaningful contribution to the field.

---

## [Author Response]

The following is the authors’ response to the original reviews

**eLife Assessment**
This important study provides the first direct neuroimaging evidence for the integration segregation theory of exogenous attention underlying inhibition of return, using an optimized IOR-Stroop fMRI paradigm to dissociate integration and segregation processes and to demonstrate that attentional orienting modulates semantic- and response-level conflict processing. Although the empirical evidence is compelling, clearer justification of the experimental logic, more cautious framing of behavioral and regional interpretations, and greater transparency in reporting and presentation are needed to strengthen the conclusions. The work will be of broad interest to researchers investigating visual attention, perception, cognitive control, and conflict processing.

We appreciate the positive reception to our manuscript. In the revised manuscript, we have further clarified the logic underlying the task design, adopted a more cautious tone in interpreting the behavioral and neuroimaging results, and enhanced the transparency of reporting and presentation.

**Public Reviews:**

**Reviewer #1 (Public review):**
Summary:This study makes a significant and timely contribution to the field of attention research. By providing the first direct neuroimaging evidence for the integration-segregation theory of exogenous attention, it fills a critical gap in our understanding of the neural mechanisms underlying inhibition of return (IOR). The authors employ a carefully optimized cue-target paradigm combined with fMRI to elegantly dissociate the neural substrates of cue-target integration from those of segregation, thereby offering compelling support for the integration-segregation account. Beyond validating a key theoretical hypothesis, the study also uncovers an interaction between spatial orienting and cognitive conflict processing, suggesting that exogenous attention modulates conflict processing at both semantic and response levels. This finding shed new light on the neural mechanisms that connect exogenous attentional orienting with cognitive control.Strengths:The experimental design is rigorous, the analyses are thorough, and the interpretation is well grounded in the literature. The manuscript is clearly written, logically structured, and addresses a theoretically important question. Overall, this is an excellent, high-impact study that advances both theoretical and neural models of attention.Weaknesses:While this study addresses an important theoretical question and presents compelling neuroimaging findings, a few additional details would help improve clarity and interpretation. Specifically, more information could be provided regarding the experimental conditions (SI and RI), the justification for the criteria used for excluding behavioral trials, and how the null condition was incorporated into the analyses. In addition, given the non-significant interaction effect in the behavioral results, the claim that the behavioral data "clearly isolated" distinct semantic and response conflict effects should be phrased more cautiously.

We thank the reviewer for these helpful comments. In the revised manuscript, we have provided additional clarification regarding the SI and RI conditions (page 29), expanded the justification for the behavioral trial exclusion criteria (page 32), and clarified how the null condition was modeled and incorporated into the analyses (page 29). In addition, we have revised the description of the behavioral results to adopt more cautious wording, particularly given the absence of a significant interaction effect. For detailed responses to these specific points, please refer to the "Recommendations for the Authors" section below.

**Reviewer #2 (Public review):**
Summary:This study provides evidence for the integration-segregation theory of an attentional effect, widely cited as inhibition of return (IOR), from a neuroimaging perspective, and explores neural interactions between IOR and cognitive conflict, showing that conflict processing is potentially modulated by attentional orienting.Strengths:The integration-segregation theory was examined in a sophisticated experimental task that also accounted for cognitive conflict processing, which is phenomenologically related to IOR but "non-spatial" by nature. This study was carefully designed and executed. The behavioral and neuroimaging data were carefully analyzed and largely well presented.Weaknesses:The rationale for the experimental design was not clearly explained in the manuscript; more specifically, why the current ER-fMRI study would disentangle integration and segregation processes was not explained. The introduction of "cognitive conflict" into the present study was not well reasoned for a non-expert reader to follow.

We thank the reviewer for raising these important points. In the revised manuscript, we have further clarified the rationale of the experimental design and the motivation for introducing cognitive conflict.

First, we clarified that previous neuroimaging studies relied primarily on SOA-based contrasts, which capture the temporal dynamics of attentional orienting but do not directly distinguish the functional processes of integration and segregation. We therefore established the direct comparison between cued and uncued targets in the long SOA as the critical test required by the theory, as these conditions are hypothesized to engage integration and segregation processes, respectively (pages 6-7, “The Challenge of Neural Verification”). Crucially, to successfully implement this comparison, we highlighted the specific methodological advantage of our study: the use of a Genetic Algorithm (GA) to optimize the stimulus sequence. We explained how this design maximizes statistical power specifically for contrast detection (i.e., cued vs. uncued) while maintaining high estimation efficiency, thereby directly overcoming the power constraints that had likely obscured these subtle neural signatures in prior ER-fMRI work (pages 7-8).

Second, we clarified that the manipulation of cognitive conflict was introduced with the additional aim of examining IOR expression mechanisms, specifically investigating how spatial attention modulates ongoing cognitive processing after target onset, rather than the generation of IOR itself. We have now provided a clearer rationale for embedding a modified Stroop task within the cue-target paradigm, and explained how this design allows us to dissociate semantic and response conflicts while avoiding methodological confounds present in previous studies (page 8).

The presentation of the results can be further improved, especially the neuroimaging results. For instance, Figure 4 is challenging to interpret. If "deactivation" (or a reduction in activation) is regarded as a neural signature of IOR, this should be clearly stated in the manuscript.

We thank the reviewer for pointing out the interpretational challenges in Figure 4. To address this, we have revised Figure 4 and provided a clearer and more precise interpretation of these interaction effects in the manuscript.

First, we have added explicit panel titles to Figure 4 (page 17). Panel A is now clearly labeled as the “Effect of IOR on Semantic Conflict”, while Panel B is labeled as the “Effect of IOR on Response Conflict”. We hope this visual labeling helps readers clearly identify the IOR modulation effects specific to each conflict type.

Second, we have revised the figure caption to explicitly define the interaction contrasts used to quantify these modulations, providing specific formulas (e.g., [UncuedRI – Uncued-SI] > [Cued-RI – Cued-SI] for response conflict) to ensure transparency.

Finally, regarding the reviewer’s comment on “deactivation”, we realized that our original figure terminology (e.g., “IOR effect under...”) might have caused confusion by mixing the interaction effect with the IOR effect itself. We have clarified that Figure 4 specifically illustrates the “Effect of IOR on the Semantic Conflict and the Response Conflict” (i.e., interaction effect between IOR and cognitive conflict). To interpret this interaction, we further examined the simple effects of conflict under each cueing condition. Specifically, we analyzed the neural signatures of semantic conflict (SI minus NE) and response conflict (RI minus SI) separately for the cued and uncued targets. Importantly, regarding the nature of the IOR effect itself (as displayed in Figure 3, page 14), it is not simply a uniform deactivation. Instead, by directly comparing the cued and uncued conditions for the neutral words, we observed neural changes in two directions: some specific regions exhibited an increased activation (Cued > Uncued), while others showed a reduced activation (Uncued > Cued). These differential patterns involved distinct brain networks and corresponded to the distinct integration and segregation mechanisms, respectively, rather than a global loss of activation (pages 20-21).

**Reviewer #3 (Public review):**
Summary:This study aims to provide the first direct neuroimaging evidence relevant to the integration-segregation theory of exogenous attention - a framework that has shaped behavioral research for more than two decades but has lacked clear neural validation. By combining an inhibition-of-return (IOR) paradigm with a modified Stroop task in an optimized event-related fMRI design, the authors examine how attentional integration and segregation processes are implemented at the neural level and how these processes interact with semantic and response conflicts. The central goal is to map the distinct neural substrates associated with integration and segregation and to clarify how IOR influences conflict processing in the brain.Strengths:The study is well-motivated, addressing a theoretically important gap in the attention literature by directly testing a long-standing behavioral framework with neuroimaging methods. The experimental approach is creative: integrating IOR with a Stroop manipulation expands the theoretical relevance of the paradigm, and the use of a genetic algorithm-optimized fMRI design ensures high efficiency. Methodologically, the study is sound, with rigorous preprocessing, appropriate modeling, and analyses that converge across multiple contrasts. The results are theoretically coherent, demonstrating plausible dissociations between integration-related activity in the fronto-parietal attention network (FEF, IPS, TPJ, dACC) and segregation-related activity in medial temporal regions (PHG, STG). The findings advance the field by supplying much-needed neural evidence for the integration-segregation framework and by clarifying how IOR modulates conflict processing.Weaknesses:Some interpretive aspects would benefit from clarification, particularly regarding the dual roles ascribed to dACC activation and the circumstances under which PHG and STG are treated as a single versus separate functional clusters. Reporting conventions are occasionally inconsistent (e.g., statistical formatting, abbreviation definitions), which may hinder readability. More detailed reporting of sample characteristics, exclusion criteria, and data-quality metrics-especially regarding the global-variance threshold-would improve transparency and reproducibility. Finally, some limitations of the study, including potential constraints on generalization, are not explicitly acknowledged and should be articulated to provide a more balanced interpretation.

We thank the reviewer for the positive and constructive assessment of our study. In response to the concerns raised, we have carefully revised the manuscript and addressed all points in detail below. In brief, we have clarified key interpretation issues in the Discussion section, including the complementary roles of dACC activation and the distinction between statistical clustering and functional interpretation of PHG and STG activations (pages 20-21). We have also improved transparency and reporting throughout the manuscript by providing more detailed sample characteristics, clarifying exclusion criteria and global variance computation, adding illustrative supplementary figures, and standardizing statistical reporting and abbreviations (pages 28, 33). Finally, we have added a concise paragraph on limitations of the study to provide a more balanced interpretation of the findings (pages 26-27). Detailed, point-by-point responses to all specific comments are provided below (see the “Recommendations for the authors” Section).

**Recommendations for the authors:**

**Reviewer #1 (Recommendations for the authors):**
Specific comments:(1) The figure caption contains an unclear sentence (lines 195-196): "The target was a 450-ms colored Chinese character presented 600 ms after the fixation cue onset at the two target locations with equal probabilities." This description is ambiguous and should be revised for clarity.

Thanks for pointing this out. In the revised manuscript, we have rephrased the figure caption to improve clarity as follows (pages 9-10):

“Each trial started with a 150-ms non-informative cue presented at one of the two peripheral boxes. After a 150-ms interstimulus interval (ISI), a 150-ms fixation cue was presented at the central fixation box. Following a further 450-ms ISI, the target, a colored Chinese character, appeared at one of the two target locations with equal probabilities and remained on the screen for 450 ms. The trial ended with a variable intertrial interval (ITI) of 850, 1050, 1250, or 1450 ms (with equal probabilities).”

(2) Please provide a more detailed and clearer description of the SI and RI experimental conditions in the Methods section.

Thanks for this helpful suggestion. We have revised the Methods section to provide a more detailed description of the SI and RI conditions. Specifically, we have further described the stimulus-response mapping and clarified how the SI and RI conditions are defined based on whether the ink color and the character meaning fell into the same or different response categories under this mapping. In addition, we have added a clarification in the Methods section to make it clearer that the SI trials involved semantic conflict without response conflict, whereas RI trials involve both semantic and response conflicts (page 29).

(3) As the data were collected across two research centers, please clarify the number of participants enrolled at each site.

Thanks for this suggestion. We have now explicitly stated in the Apparatus and Data Acquisition section that 16 participants were enrolled at each site. The revised text reads (page 31)：

“The imaging data were acquired at two research sites following comparable protocols, with equal numbers of participants scanned at each site (n = 16 per site).”

(4) In the behavioral data analysis, please provide the rationale or justification for the criteria used to exclude trials.

Thanks for this comment. In the revised manuscript (page 32), we have clarified that reaction times (RTs) shorter than 150 ms were excluded as anticipatory responses, and RTs longer than 1,300 ms were excluded to limit the influence of unusually slow responses. These exclusion criteria are commonly adopted in RT research and were applied consistently across all conditions (Ratcliff, 1993; Whelan, 2008).

(5) Given that the behavioral interaction effect was not statistically significant, the conclusion on lines 236-237, "These data clearly isolated the two distinct conflict effects in the Stroop effect, namely the semantic conflict (SI-NE difference) and the response conflict (RI-SI difference)" appears overstated and should be softened accordingly.

We thank the reviewer for this important comment. We have clarified that our original statement was intended to highlight the successful isolation of conflict types based on the significant main effects of congruency (validating the task design), rather than implying a significant interaction effect. However, we agree that the original phrasing appeared unclear in this context. We have therefore revised the sentence to adopt a more cautious tone in the revised manuscript (page 12):

“These data demonstrated typical Stroop interference effects (Veen & Carter, 2005) in both the semantic (SI-NE difference) and response conflicts (RI-SI difference).”

(6) The statement on lines 281-282, "Although the IOR effect showed no effect on either the semantic conflict difference (SI-NE) or the response conflict difference (RI-SI) in the behavioral performance" lacks supporting statistical evidence. Please report the relevant test statistics.

We appreciate the reviewer’s careful reading and note that the relevant statistical evidence was missing from the original manuscript. This has now been added in the revised version. Specifically, we examined the interactions between cue validity and semantic conflict (SI vs. NE) as well as between cue validity and response conflict (RI vs. SI). Neither interaction was significant (see revised Results for full statistics on page 12), supporting our original statement that cue validity did not modulate either conflict component in behavioral performance.

(7) The manuscript mentions that a null condition (with no Chinese character presented) was included to increase statistical power for detecting differences across conditions. However, it is unclear how this null condition was actually used in the data analyses. Please clarify the role of the null condition in both the behavioral and neuroimaging analyses.

Thanks for this comment. We regret that this was not sufficiently clear in the original manuscript. The null condition was included for neuroimaging purposes and was not used in the behavioral analyses, as no response was required in these trials. In the fMRI analyses, null trials served as the implicit baseline and were not modeled as regressors of interest. Task-related activities for all experimental conditions were therefore estimated relative to this null baseline, facilitating estimations of task-related responses in randomized event-related designs (Burock et al., 1998; Friston et al., 1999; Liu, 2004). We have clarified this point in the revised manuscript (page 29).

References

Burock, M. A., Buckner, R. L., Woldorff, M. G., Rosen, B. R., & Dale, A. M. (1998). Randomized event-related experimental designs allow for extremely rapid presentation rates using functional MRI. NeuroReport, 9(16), 3735-3739. https://doi.org/10.1097/00001756-199811160-00030

Friston, K. J., Zarahn, E., Josephs, O., Henson, R. N. A., & Dale, A. M. (1999). Stochastic designs in event-related fMRI. NeuroImage, 10(5), 607-619. https://doi.org/10.1006/nimg.1999.0498

Liu, T. T. (2004). Efficiency, power, and entropy in event-related fMRI with multiple trial types: Part II: design of experiments. NeuroImage, 21(1), 401-413. https://doi.org/10.1016/j.neuroimage.2003.09.031

Ratcliff, R. (1993). Methods for dealing with reaction time outliers. Psychological Bulletin, 114(3), 510-532. https://doi.org/10.1037/0033-2909.114.3.510

Whelan, R. (2008). Effective analysis of reaction time data. The Psychological Record, 58(3), 475-482. https://doi.org/10.1007/BF03395630

**Reviewer #2 (Recommendations for the authors):**
(1) The paper is a bit too lengthy, with a lot of information that is hard for non-experts to grasp.

We thank the reviewer for this comment. We realized that the Introduction was the most challenging section for general readers. In the revision, we refined the text in the Introduction for a better structure and more reader-friendly wording to improve readability. In addition, following the reviewer’s suggestion (Recommendation 4 below), we have added short subsection titles to the Introduction, Results, and Discussion sections to better organize the content and highlight the main ideas. We hope these revisions make the manuscript more accessible and easier for a broader audience to follow.

(2) Please double-check the stats, as some of the results presented in the main text do not align well with the figures. Take Figure 2 as an example.

We appreciate the reviewer’s concern and have double-checked all statistics. All the results are consistent between the figures and the main text. Take Figure 2 as an example (page 12), the perceived discrepancy probably was caused by the fact that the descriptive values reported in the main text are marginal means for the main effects (i.e., the overall average of one factor, collapsed over the other factor), whereas Figure 2 shows the mean for each Congruency × Cue Validity condition (i.e., simple effect).

(3) The reasoning that the neuroimaging findings support the dissociation between integration and segregation needs to be improved.

We thank the reviewer for this important comment. In the revised Discussion (pages 1921), we have strengthened the reasoning linking our neuroimaging findings to the dissociation between the integration and segregation processes. Specifically, we make it clear how the distinct activation patterns observed for the cued and uncued targets map onto the different functional demands proposed by the integration-segregation theory. The cued targets were theorized to recruit the frontoparietal attentional control networks, consistent with the re-engagement of an existing object file (integration). On the other hand, the uncued targets should engage the medial temporal and temporal association regions responsible for novelty detection and episodic encoding, consistent with the creation of a new object file (segregation). We hope the reviewer finds that the revision offers a clearer explanation of how the observed neural patterns are consistent with a dissociation between the integration and segregation processes.

(4) Please use short section titles to organize the introduction, results, and discussion sections. For instance, the discussion section is a long chunk of text (almost 9 pages) and is pretty dense, making it hard to quickly grasp the ideas the authors want to convey.

Thanks for this helpful suggestion. Following the reviewer’s recommendation, we have now added short subsection titles to the Introduction and Discussion sections to improve structure and readability. For the Results section, we have maintained and further refined the existing subheadings to ensure consistent organization.

**Reviewer #3 (Recommendations for the authors):**
I found this manuscript to be a timely and substantive contribution to the study of attention and cognitive neuroscience. To my knowledge, it provides the first direct neuroimaging evidence relevant to the integration-segregation theory of exogenous attention, a framework that has been influential in behavioral work for more than two decades but has lacked clear neural support. The study is conceptually well motivated, methodologically solid, and generally clearly reported. The findings differentiate neural substrates associated with integration and segregation processes and further show how inhibition of return (IOR) interacts with semantic and response conflicts at the neural level.The manuscript is well organized, the writing is mostly clear, and the progression from theory to hypotheses and methods is easy to follow. The combination of IOR with a modified Stroop paradigm is a clever choice that extends the theoretical scope of exogenous attention research. The use of an optimized event-related fMRI design based on a genetic algorithm is also a strength and reflects careful attention to design efficiency.The main results are internally consistent and theoretically meaningful. Integration related activity in the fronto-parietal attention network (including FEF, IPS, TPJ, and dACC) and segregation-related activity in medial temporal areas (PHG and STG) it well with the proposed framework, and the pattern of activations is coherent across analyses.Overall, I think this is a carefully executed study that offers much-needed neural evidence bearing on the integration-segregation theory of exogenous attention. I would recommend the following revisions.Suggestions:(1) In the Discussion (pp. ~17-18), dACC activation is described both in terms of general cognitive control demands and as reflecting a possible inhibitory bias toward the cued direction. It would help the reader if you could briefly indicate whether you see these as complementary (e.g., dual roles within the same region) or as more competing interpretations.

We thank the reviewer for this helpful comment. We have clarified in the revised manuscript that dACC exerts general cognitive control demands and biasing against the cued direction are complementary rather than competing interpretations. Specifically, we described how the dACC is involved in both the cognitive control required for target integration and the inhibitory bias toward the cued location, thereby highlighting its dual roles within the same region. The revised section reads as follows (page 20):

“Furthermore, the observed increase in the left dACC activity under the cued relative to the uncued condition likely reflected the engagement of cognitive control mechanisms (Botvinick et al., 2004; Chung et al., 2024; Mayer et al., 2012; Veen & Carter, 2005), particularly in resolving the conflict between the task-driven requirement of target integration and the reduced accessibility of the cue-initiated representation. In this context, the heightened activation of dACC may also reflect its role in fulfilling the inhibitory bias toward the cued location (Mayer et al., 2004) and discouraging inefficient integration attempts at a location marked as less relevant.”

(2) In the Discussion, you could consider adding a short paragraph explicitly acknowledging a few limitations and how they might constrain generalization of the findings. A concise reflection of this kind would give a more balanced picture without undermining the main conclusions.

We appreciate this helpful suggestion. In the revised manuscript, we have added a concise paragraph explicitly addressing a key limitation of the present study (pages 26-27). Specifically, we acknowledge that the absence of behavioral interactions alongside clear neural effects requires cautious interpretation. We discussed how this dissociation may reflect differences in measurement sensitivity between behavioral and neural indices, consistent with prior findings (Chen et al., 2006; Wilkinson & Halligan, 2004). We also note that the use of a GA-optimized sequence, while improving statistical efficiency, may have introduced unintended regularities in event order that could influence behavioral strategies.

(3) Since the dataset is hosted on GitHub, adding a short note in the Data Availability section about whether the repository will also include analysis scripts or future replication data would further enhance transparency and long-term usefulness.

Thanks for this helpful suggestion. We have revised the Data Availability section (page 35) to clarify that the GitHub repository contains the processed data used in the final analyses. Analysis scripts and additional materials for replication are available from the authors upon reasonable request.

(4) In the Results section, the formatting of statistics is not fully consistent. For example, some reports use spaces around symbols (e.g., "η^2^ = 0.301") whereas others do not (e.g., "p< .001"). It would be good to standardize this (e.g., "p < .001", "η^2^ = .30") across the manuscript.

Done as suggested.

(5) A few abbreviations appear before they are defined-for instance, SPC (superior parietal cortex) shows up in the Results (response conflict section) before the full name is given. Ensuring that each abbreviation is defined at first mention would help readers who may be less familiar with all of the regional acronyms.

Thanks for this comment. We have conducted a thorough check of the manuscript and ensured that all abbreviations are defined upon their first occurrence.

(6) The text sometimes refers to "PHG/STG" as a combined cluster, while at other points, PHG and STG are described separately. It would be useful to clarify under what circumstances they are treated as a single functional cluster versus distinct regions of interest, and to keep the nomenclature as consistent as possible between the main text and the tables.

Thanks for raising this point. In the revised manuscript, we have clarified this issue by distinguishing between statistical clustering and functional interpretation. In the whole brain analysis, activations in the left hemisphere formed a single continuous cluster spanning the PHG and STG; therefore, this cluster is labeled as “PHG/STG” in Table 1. We have explicitly noted the continuous nature of this cluster in the Results section (page 15) to ensure clarity:

“Notably, in the left hemisphere, these activations formed a continuous cluster spanning both regions (labeled as PHG/STG in Table 1).”

(7) It would be helpful to provide a bit more detail about the sample characteristics (e.g., age range, handedness, and inclusion/exclusion criteria) and to state explicitly how many participants, if any, were excluded from the analyses and for what reasons. This would help readers better evaluate data quality and generalizability.

Thanks for this helpful suggestion. We have revised the Participants section (page 28) to provide the full details regarding our sample:

“32 healthy participants with normal or corrected-to-normal vision and normal color vision were recruited. All participants were right-handed and reported no history of neurological or psychiatric disorders. Data from three participants were excluded due to excessive head movements and high global variances (see fMRI Data Analysis), leaving 29 participants for analysis (18 female, 11 male; aged 18-30 years, M = 22.69, SD = 2.58).”

Furthermore, we have provided a clearer description of the exclusion criteria in the Data Analysis section (pages 33-34) as follows:

“Runs with motions exceeding one voxel length in any direction were excluded (resulting in the exclusion of two runs) …Runs with global variance equal to or over 0.1% were excluded, resulting in the exclusion of eight runs (see Supplementary Information for details). Ultimately, three participants were excluded because neither run met the quality criteria. All remaining participants retained both runs, except for three individuals who each contributed only one valid run.”

(8) Given that participants were excluded based on global variance exceeding 0.1%, it would be very informative to include, in the Supplementary Materials, an illustrative figure showing the signal time series (or global signal variance over time) for excluded participants.

We appreciate this valuable suggestion. In the revised Supplementary Materials, we have included a new figure (Figure S2) that plots the global signal time series for the excluded runs to illustrate the signal patterns that led to their exclusion based on global variance.

(9) Relatedly, it may help to more explicitly describe how global variance was computed (e.g., over which time window, after which preprocessing steps, and whether it was calculated on whole-brain signal or within specific masks). A concise clarification would make the exclusion criterion easier to interpret.

Thanks for this helpful suggestion. We have now clarified in the manuscript how global variance was computed (page 33) and have also provided a more detailed description of the computation procedure in the Supplementary Materials (page 4). Specifically, after the standard preprocessing (slice timing correction, 3D motion correction, spatial smoothing, linear trend removal, and high-pass temporal filtering), the global signal was computed for each run as the mean signal across voxels with intensity values greater than 100 in each volume. Global variance was then quantified as the temporal variance of this run-wise global-signal time course across all volumes, providing a quality-control index of signal stability.

(10) Rather than only reporting a single overall exclusion rate (e.g., 5.52% of total trials), it would be informative to break this down by source, reporting separately the proportion of trials excluded as RT outliers and the proportion excluded due to response errors. This would further improve transparency regarding the behavioral preprocessing pipeline.

Thanks for this helpful suggestion. We have now broken down the overall exclusion rate by source in the revised manuscript. Specifically, we reported that 4.29% of trials were excluded due to incorrect responses, and 1.24% of trials were excluded as RT outliers (page 32).

References

Botvinick, M. M., Cohen, J. D., & Carter, C. S. (2004). Conflict monitoring and anterior cingulate cortex: an update. Trends in Cognitive Sciences, 8(12), 539-546. https://doi.org/10.1016/j.tics.2004.10.003

Chen, Q., Wei, P., & Zhou, X. (2006). Distinct neural correlates for resolving stroop conflict at inhibited and noninhibited locations in inhibition of return. Journal Of Cognitive Neuroscience, 18(11), 1937-1946. https://doi.org/10.1162/jocn.2006.18.11.1937

Chung, R. S., Cavaleri, J., Sundaram, S., Gilbert, Z. D., Del Campo-Vera, R. M., Leonor, A., Tang, A. M., Chen, K.-H., Sebastian, R., Shao, A., Kammen, A., Tabarsi, E., Gogia, A. S., Mason, X., Heck, C., Liu, C. Y., Kellis, S. S., & Lee, B. (2024). Understanding the human conflict processing network: A review of the literature on direct neural recordings during performance of a modified stroop task. Neuroscience Research, 206, 1-19. https://doi.org/10.1016/j.neures.2024.03.006

Mayer, A. R., Seidenberg, M., Dorflinger, J. M., & Rao, S. M. (2004). An event-related fMRI study of exogenous orienting: supporting evidence for the cortical basis of inhibition of return? Journal Of Cognitive Neuroscience, 16(7), 1262-1271. https://doi.org/10.1162/0898929041920531

Mayer, A. R., Teshiba, T. M., Franco, A. R., Ling, J., Shane, M. S., Stephen, J. M., & Jung, R. E. (2012). Modeling conflict and error in the medial frontal cortex. Human Brain Mapping, 33(12), 2843-2855. https://doi.org/10.1002/hbm.21405

Veen, V. V., & Carter, C. S. (2005). Separating semantic conflict and response conflict in the Stroop task: A functional MRI study. Neuro Image, 27(3), 497-504. https://doi.org/10.1016/j.neuroimage.2005.04.042

Wilkinson, D., & Halligan, P. (2004). The relevance of behavioural measures for functional imaging studies of cognition. Nature Reviews Neuroscience, 5(1), 67-73. https://doi.org/10.1038/nrn1302